# Recruitment of the mitotic exit network to yeast centrosomes couples septin displacement to actomyosin constriction

Davide Tamborrini[1,3], Maria Angeles Juanes [1,4], Sandy Ibanes[1], Giulia Rancati[2] & Simonetta Piatti[1]

In many eukaryotic organisms cytokinesis is driven by a contractile actomyosin ring (CAR) that guides membrane invagination. What triggers CAR constriction at a precise time of the cell cycle is a fundamental question. In budding yeast CAR is assembled via a septin scaffold at the division site. A Hippo-like kinase cascade, the Mitotic Exit Network (MEN), promotes mitotic exit and cytokinesis, but whether and how these two processes are independently controlled by MEN is poorly understood. Here we show that a critical function of MEN is to promote displacement of the septin ring from the division site, which in turn is essential for CAR constriction. This is independent of MEN control over mitotic exit and involves recruitment of MEN components to the spindle pole body (SPB). Ubiquitination of the SPB scaffold Nud1 inhibits MEN signaling at the end of mitosis and prevents septin ring splitting, thus silencing the cytokinetic machinery.

[1] Centre de Recherche en Biologie Cellulaire de Montpellier (CRBM), 1919 Route de Mende, 34293 Montpellier, France. [2] Institute of Medical Biology, 8a Biomedical Grove, Singapore 138648, Singapore. [3] Present address: Max-Planck-Institute of Molecular Physiology, Otto-Hahn Str. 11, 44227 Dortmund, Germany. [4] Present address: Brandeis University, 415 South Street, Waltham, MA 02454, USA. Correspondence and requests for materials should be addressed to S.P. (email: simonetta.piatti@crbm.cnrs.fr)

Cytokinesis is the final stage of mitosis leading to the physical separation of the two daughter cells. In many eukaryotic organisms, such as fungi and animals, cytokinesis is driven by a contractile actomyosin ring (CAR) at the site of cell division. CAR constriction during cytokinesis drives invagination of the overlying plasma membrane inward to cleave the cell in two. Besides generating force, CAR constriction in yeast is also coupled to membrane deposition and formation of a primary septum[1,2].

Septins have been implicated, besides CAR, in cytokinesis in many eukaryotes. Septins are cytoskeletal guanosine triphosphate (GTP)-binding proteins that form oligomeric complexes that can in turn self-organize in higher-order structures, such as filaments and rings. Studies in budding yeast and mammalian cells indicate that septins act as scaffolds to recruit cytokinesis factors to the site of cell division and regulate CAR constriction (reviewed in ref. [3]). Furthermore, *Drosophila* and human (but not yeast) septins bundle and bend actin filaments for CAR assembly[4].

Septins are essential for cytokinesis in the budding yeast *Saccharomyces cerevisiae*, where they recruit CAR components and other cytokinetic proteins to the division site (reviewed in ref. [3]). Budding yeast septins form rod-shaped heteromeric complexes that join end-to-end in nonpolar filaments, which in turn organize in a ring that interacts tightly with the plasma membrane at the bud neck, the constriction between mother and future daughter cell[5,6]. Septins are first recruited in the G1 phase of the cell cycle to the presumptive bud site as unorganized septin clouds or patches, which are then rapidly transformed into a cortical septin ring. At the time of bud emergence the septin ring expands into an hourglass-shaped septin collar, which spans the whole bud neck. Immediately prior to cytokinesis the septin collar suddenly splits into two distinct rings that sandwich the constricting CAR[7,8]. This remarkable rearrangement is accompanied by a 90° rotation of septin filaments, which are aligned parallel to the mother-bud axis in the collar while they lie orthogonally to it in the split rings[9]. Furthermore, fluorescence recovery after photobleaching experiments showed that while the septin collar is a rigid structure, split septin rings are dynamic[10,11]. The relevance of septin ring splitting for cytokinesis is poorly understood, mainly due to the lack of mutants specifically defective in this process. Since both septins and the CAR must contact the plasma membrane, it is plausible that septins impose a physical constraint to CAR assembly or contraction that is overcome by septin splitting. However, this hypothesis could not be experimentally tested so far.

The mitotic exit network (MEN) is an essential Hippo-like kinase cascade that promotes mitotic exit and cytokinesis in budding yeast (reviewed in ref. [12]). MEN includes the upstream GTPase Tem1, which activates the Ste20-like Cdc15 kinase that in turn upregulates the NDR kinases Dbf2 and Dbf20 in association with their Mob1 activator. The Tem1 GTPase can be inhibited by the two component GTPase-activating protein (GAP) Bub2-Bfa1[13], whose activity is antagonized by the polo kinase Cdc5 through Bfa1 phosphorylation[14]. Many MEN factors localize in a cell cycle-regulated manner at the yeast centrosome, called spindle pole body (SPB). Their recruitment to SPBs is mediated by the centriolin-related scaffold protein Nud1 and is crucial for MEN signaling[15–19]. The final target of MEN is the Cdc14 phosphatase, which is trapped in the nucleolus in an inactive state from G1 to anaphase and then released in the nucleoplasm and cytoplasm by MEN signaling. In turn, Cdc14 brings about mitotic exit by inactivating mitotic CDKs and reversing phosphorylations of CDK substrates (reviewed in ref. [20]). Although the latter is a critical prerequisite for licensing cytokinesis in many organisms, MEN factors promote cytokinesis also independently of mitotic exit (reviewed in ref. [12]). In fission yeast a Hippo-like signaling cascade, called septation initiation network (SIN), has exactly the same organization of MEN and is essential for cytokinesis without being involved in mitotic exit (reviewed in ref. [21]).

The MEN GTPase Tem1 was shown to promote both septin ring splitting and CAR contraction independently of Cdc14 release from the nucleolus[7], raising the possibility that the two processes are coupled. Knowing that CAR components are dispensable for septin splitting[7], whether Tem1 promotes solely septin ring splitting, thereby indirectly promoting CAR contraction, or controls both processes separately is a key question that remains to be addressed. Similarly, how Tem1 controls septin splitting has yet to be investigated.

Taking advantage of yeast strains that are specifically defective in septin ring splitting, we demonstrate that septin ring splitting/displacement is an essential prerequisite for CAR contraction and for cytokinesis. Furthermore, we show that MEN signaling at SPBs is crucial for this process through recruitment of the Cdc14 phosphatase to SPBs, but independently of its involvement in mitotic exit. Ubiquitination of the MEN scaffold Nud1 at SPBs silences septin splitting and CAR contraction once these processes have occurred. Altogether, our data highlight the importance of a critical cytokinetic step that is likely conserved in other eukaryotic systems.

## Results

**Septin ring splitting and AMR contraction are spatially and temporally separated**. The myosin II Myo1, which is a major CAR component[22,23], is first recruited to the septin ring in late G1 and forms the CAR in late mitosis[24]. To determine if the contractile Myo1 ring is still connected to septins after their splitting, we applied super-resolution three-dimensional structured illumination microscopy (3D-SIM) on fixed cells expressing the septin Shs1 tagged with mCherry along with GFP-tagged Myo1. We found that the Myo1 ring has a smaller diameter than the split septin rings (0.6 vs. 1 μm) and it is placed 0.2 μm away from the split septin rings (Fig. 1a). Thus, at the time of cytokinesis CAR and septins are physically separated.

Previous data showed that CAR constriction occurs approximately at the same time as septin ring splitting[7,8]. However, the exact timing between the two events has not been determined. We therefore carefully quantified the fluorescence associated to Shs1-mCherry and Myo1-GFP at the bud neck during cytokinesis by live cell imaging. Indeed, septin ring splitting is accompanied by loss of septin subunits, which causes a decrease in Shs1 fluorescence[8]. Additionally, the relative density of Myo1 at the CAR remains constant during contraction, decreasing in levels while CAR circumference shrinks[22,23]. Our measurements indicate that septin ring splitting precedes by 4–5 min CAR contraction (Fig. 1b). We conclude that the two events are spatially and temporally separated.

**MEN factors are required for septin ring splitting independently of mitotic exit**. To get a comprehensive view of the control of septin ring splitting and CAR constriction by the MEN cascade (Supplementary Fig. 1g), we analyzed these events by time lapse imaging in conditional MEN mutants expressing either wild-type *CDC14* or the dominant *CDC14^TAB6-1* allele that partially bypasses MEN requirement for mitotic exit by loosening Cdc14 association with its nucleolar anchor[25]. As expected, the temperature-sensitive *nud1-44*, *dbf2-2*, *mob1-77*, *cdc14-3*, as well as the repressible *GAL1-UPL-TEM1* and the analogue-sensitive *cdc15-as1* mutants, in restrictive conditions arrested in late mitosis with large buds, unsplit septin rings and stable CAR at the bud neck (Supplementary Fig. 1a–f). In agreement with previous data[7], forcing mitotic exit in *GAL1-UPL-TEM1* cells through the

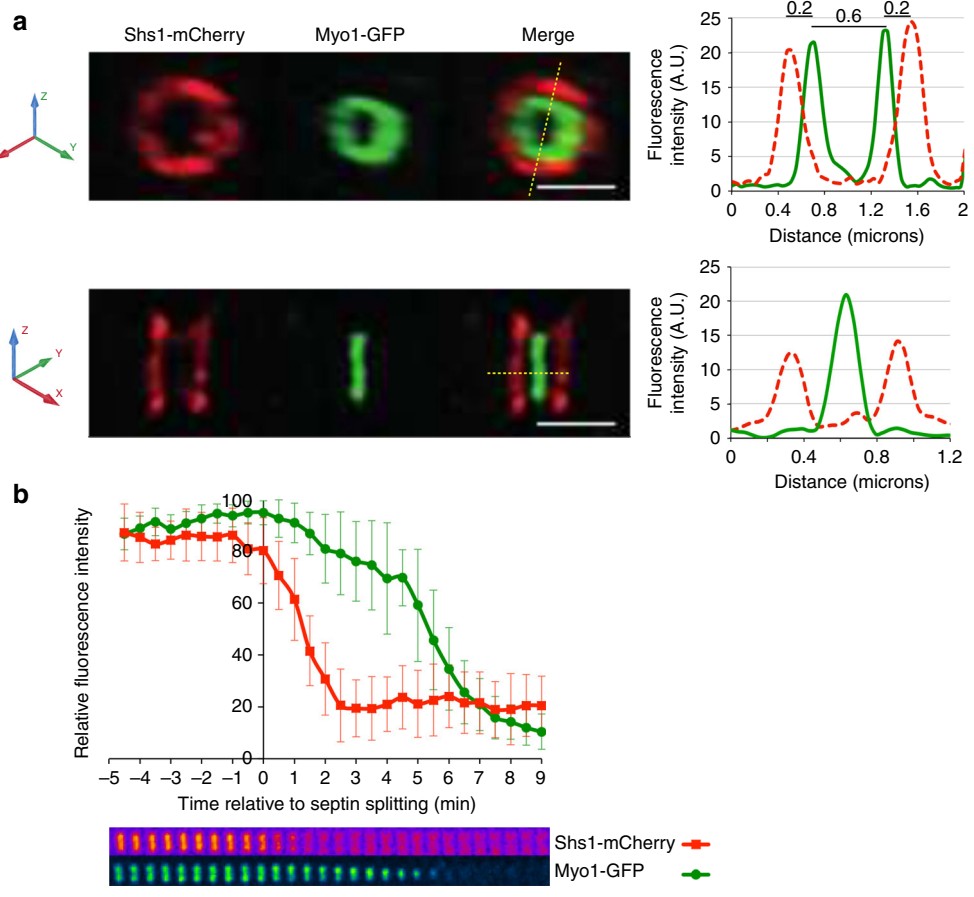

**Fig. 1** Septin ring splitting and CAR constriction are spatially and temporally separated events. **a** Logarithmically growing cells expressing Shs1-mCherry and Myo1-GFP were fixed and processed for SIM. The image shows an example of split septin rings sandwiching the CAR. Scale bar: 2 μm. Graphs show the quantification of fluorescence intensities along the yellow dotted line in the merge. Dotted red line: Shs1-mCherry; green line: Myo1-GFP. A.U.: Arbitrary Units. **b** Same cells as in **a** were imaged live every min through their cell cycle. Quantification of fluorescence intensities associated to Shs1-mCherry and Myo1-GFP around the time of septin ring splitting (time 0). Fluorescence intensity associated to septin and myosin II has been quantified by ImageJ in cells undergoing cytokinesis (graph; red squares: Shs1-mCherry; green circles: Myo1-GFP) and then related to the highest fluorescence intensity of each structure in a given cell. Plots show average values ($n = 15$). Error bars: s.d. Cropped images beneath the graph show the behavior of septin ring and CAR during this time frame in one representative cell. Shs1 was pseudocolored with the Fire plugin of Image J to reflect signal intensity (orange/red signals mean higher fluorescence intensity than magenta signals)

$CDC14^{TAB6-1}$ allele allowed entry into a new cell cycle without cytokinesis, as assessed by rebudding in the absence of septin ring splitting or CAR constriction (Fig. 2a). Furthermore, fluorescence-activated cell sorting (FACS) analysis on synchronized cell populations showed that while $GAL1$-$UPL$-$TEM1$ cells arrested mainly with 2C DNA content, $GAL1$-$UPL$-$TEM1$ $CDC14^{TAB6-1}$ cells exited mitosis and underwent a second round of DNA replication without cytokinesis, as shown by the accumulation of cells with 4C DNA content (Fig. 2b).

We then asked which MEN components are required for septin ring splitting downstream of Tem1. Similar to Tem1 inactivation, inhibition of the Tem1 effector kinase Cdc15 in $cdc15$-$as1$ $CDC14^{TAB6-1}$ cells prevented both septin ring splitting and CAR constriction (Fig. 2c). This resulted in prominent cytokinesis defects, as shown by FACS analysis of DNA contents on whole cell populations (Fig. 2d).

Cdc15 activates the downstream Dbf2 kinase in association with its activating subunit Mob1, both through Dbf2 phosphorylation and recruitment of the Mob1–Dbf2 complex to SPBs by phosphorylation of the scaffold protein Nud1[16,26]. Mob1 inactivation through the temperature-sensitive $mob1$-$77$ allele in combination with $CDC14^{TAB6-1}$ led to pronounced cell lysis in most cells in synthetic medium (SD) medium at 32 and 34 °C.

However, in a few cells that remained intact during the temperature shift we could observe mitotic exit without concomitant septin ring splitting and CAR constriction (Fig. 2e), consistent with previously reported cytokinesis defects[27]. These were further confirmed by FACS analysis of DNA contents on synchronized cells populations (Fig. 2f). In sharp contrast, inactivation of the Dbf2 kinase through the temperature-sensitive $dbf2$-$2$ allele in $CDC14^{TAB6-1}$ cells did not prevent either septin splitting or CAR constriction (Supplementary Fig. 2a), allowing cytokinesis in virtually all cells at 34 °C (Supplementary Fig. 2b). Similar results were obtained by additionally deleting the Dbf2 paralogue Dbf20 in $dbf2$-$2$ $CDC14^{TAB6-1}$ cells at 35.5 °C, i.e., the maximal temperature at which these cells could still exit mitosis (Supplementary Fig. 2c).

To definitely ascertain if Dbf2 is dispensable for septin ring splitting, we introduced one or three miniAID tags (AID: auxin-inducible degron[28]) at the 3′ end of the $dbf2$-$2$ open reading frame to allow for the rapid depletion of Dbf2 in the presence of indoleacetic acid (IAA) and upon expression of the E3 ligase OsTir1 from the galactose-inducible $GAL1$ promoter. Insertion of $1miniAID$ or $3miniAID$ at the 3′ end of $dbf2$-$2$ was lethal in $dbf20Δ$ cells even in the absence of IAA and galactose, suggesting that tagging compromises Dbf2 protein function. In contrast,

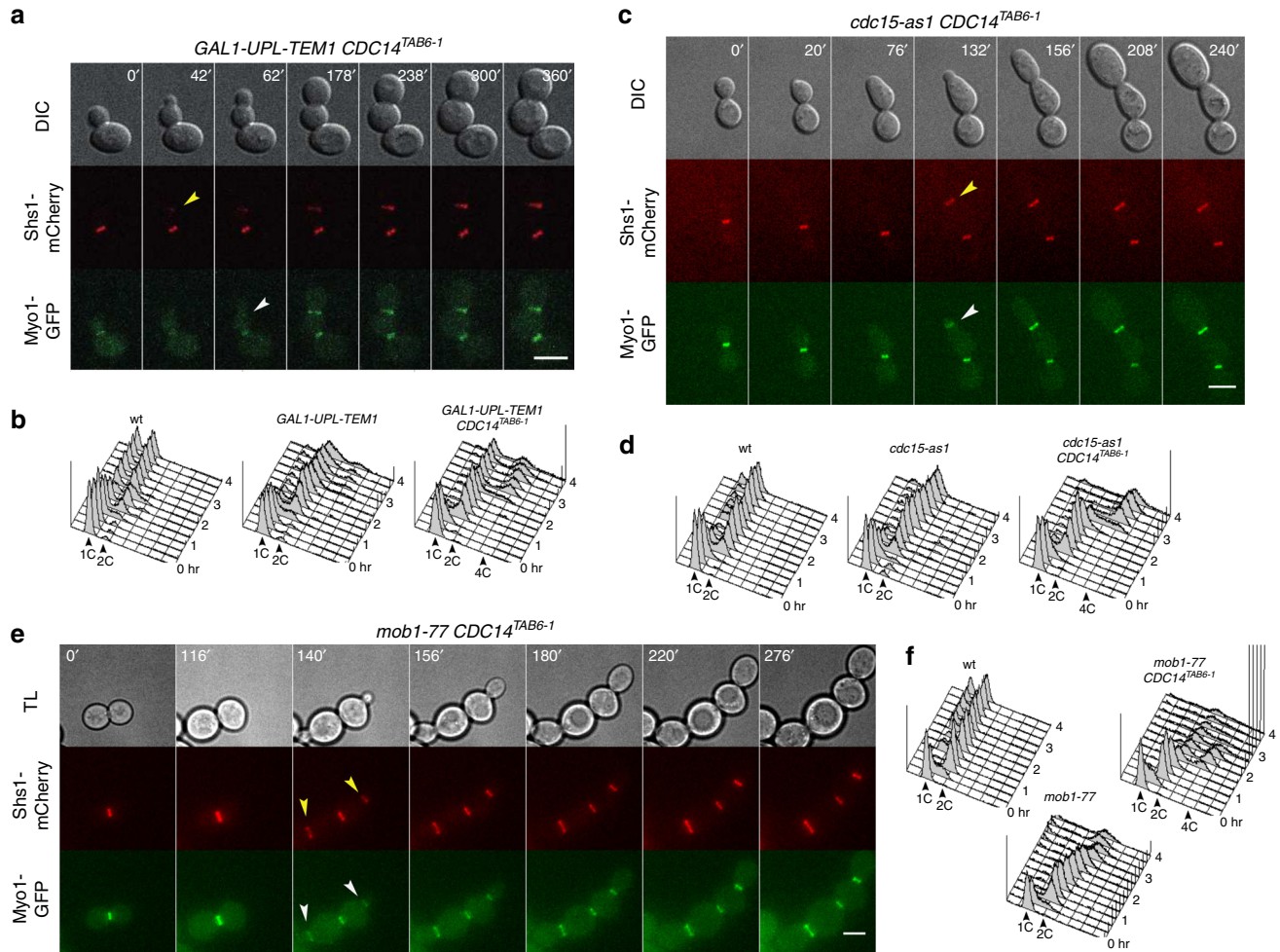

**Fig. 2** The MEN factors Tem1, Cdc15, and Mob1 are required for septin ring splitting and CAR contraction independently of mitotic exit. **a**, **c**, **e** Cells with the indicated genotypes were grown in permissive conditions and then shifted to restrictive conditions 60–90 min prior to imaging. Cells were filmed every 2 min (**a**) or 4 min (**c**, **e**) for 4–8 h in restrictive conditions (**a** glucose-containing medium; **c** medium supplemented with 5 μM 1NM-PP1; **e** 32 °C). Arrowheads indicate the appearance of new septin rings (yellow) or CARs (white) before the old structures have been disassembled. DIC differential interference contrast. TL transmitted light. Scale bar: 5 μm. **b**, **d**, **f** Cells with the indicated genotypes were grown in permissive conditions (**b** YEPRG; **d**, **f** YEPD) at 25 °C, arrested in G1 with alpha factor and then released in restrictive conditions (**b** YEPD; **d** YEPD containing 5 μM 1NM-PP1; **f** YEPD at 32 °C). At various time points after release (time 0) cells were collected for FACS analysis of DNA contents. FACS data were plotted after gating out the debris as illustrated in Supplementary Fig. 12

$dbf20\Delta$ $CDC14^{TAB6-1}$ cells carrying $dbf2$-$2$-$miniAID$ constructs were viable and proliferated efficiently in glucose- and galactose-containing medium ($GAL1$-$OsTIR1$ off and on, respectively; Supplementary Fig. 2f), indicating that entrapment of Cdc14 in the nucleolus is the main cause of the lethality linked to AID-tagging of $dbf2$-$2$. Furthermore, $dbf2$-$2$-$3miniAID$ $dbf20\Delta$ $CDC14^{TAB6-1}$ $GAL1$-$OsTIR1$ cells stopped proliferating on IAA-containing galactose medium at 30 °C (Supplementary Fig. 2f), indicating that Dbf2 depletion could be efficiently achieved. Imaging of the septin GFP-Cdc12 in these cells dividing in the presence of IAA and galactose at 30 °C confirmed that the Dbf2/Dbf20 kinases are not required for septin ring splitting (Supplementary Fig. 2e), in agreement with previous conclusions[29,30]. Indeed, all cells that exited mitosis during the movie, as assessed by the appearance of a new bud and a new septin ring, previously split the pre-existing septin ring at the bud neck ($n = 53$).

Thus, the whole MEN cascade is essential for septin ring splitting and CAR constriction through the downstream Cdc14 phosphatase. Additionally, the Tem1 GTPase, its effector kinase Cdc15 and the Mob1 protein, but not its associated kinases Dbf2/

Dbf20, are required for these processes also independently of their role in mitotic exit.

**The ubiquitin-ligase Dma2 prevents septin ring splitting and CAR constriction**. We previously showed that overexpression of the E3 ubiquitin ligase Dma2 prevents septin ring splitting and cytokinesis without hampering mitotic exit, thus causing the accumulation of chains of cells with stable septin rings at bud necks and accumulation of ≥4C DNA contents[31,32] (Fig. 3a). We, therefore, wondered if lack of septin ring splitting was accompanied by a failure to constrict the CAR. Time lapse imaging of cells overexpressing $DMA2$ from the galactose-inducible $GAL1$ promoter and expressing Shs1-mCherry along with Myo1-GFP showed indeed that CAR was not contracting. At the end of the cell cycle, cells exited mitosis and rebudded after forming a new septin ring, but kept the old septin collar and unconstricted CAR at the bud neck (Fig. 3b). This prevented formation of a septum between the two dividing cells that in most cases shared a common cytoplasm, as shown by transmission electron microscopy (Fig. 3c). Introducing the $CDC14^{TAB6-1}$ allele in $GAL1$-$DMA2$

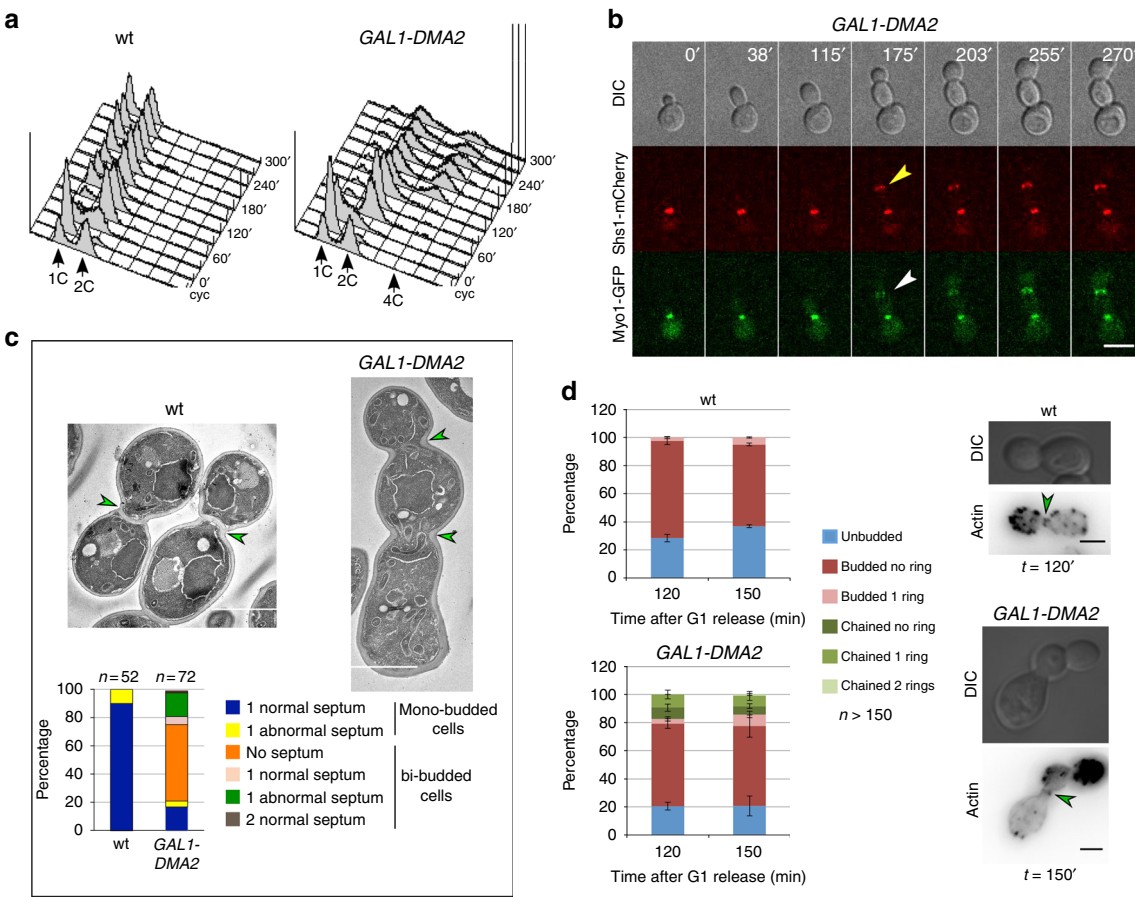

**Fig. 3** Overexpression of the E3 ubiquitin ligase Dma2 prevents septin ring splitting, CAR contraction and cytokinesis. **a** Wild-type and *GAL1-DMA2 bud4-G820fs* cells were grown in YEPR at 25 °C, arrested in G1 by alpha factor and induced with 1% galactose 30 min before the release. Cells were finally released in YEPRG at 30 °C (time 0). Cells were collected at the indicated time points for FACS analysis of DNA contents. FACS data were plotted after gating out the debris as illustrated in Supplementary Fig. 12. **b** *GAL1-DMA2 BUD4* cells expressing Shs1-mCherry and Myo1-GFP grown in SD-raffinose were induced for 90 min with galactose and then imaged in SD-raffinose/galactose at 30 °C. Arrowheads indicate the appearance of new septin rings (yellow) or CARs (white) before the old structures have been disassembled. DIC: differential interference contrast. Scale bar: 5 μm. **c** Wild-type and *GAL1-DMA2 bud4-G820fs* cells were treated as in **a**. At 240 min after release cells were fixed and processed for transmission electron microscopy. Scale bar: 2 μm. **d** Wild-type and *GAL1-DMA2 BUD4* cells were treated as in **a**. At the indicated times after release cells were fixed for phalloidin staining of actin structures. Data are means from three independent experiments. Error bars: s.d. Micrographs show representative cells

cells did not improve their ability to split septin rings or to constrict the CAR (Fig. 4e). These data confirm that *DMA2* overexpression interferes with, without blocking, some aspects of mitotic exit[31]. Consistently, the chitin synthase Chs2, which gets recruited to the bud neck at the onset of cytokinesis by MEN-dependent activation of the Cdc14 phosphatase[2,33], did not appear at the division site of *GAL1-DMA2* cells that failed to undergo septin splitting (Supplementary Fig. 3a, b, d).

Since we recently showed that Dma1/2 control the localization of the formins Bni1 and Bnr1 at polarity sites[34], which in turn is important for CAR assembly[35], we asked if F-actin was timely recruited to the CAR in Dma2-overexpressing cells. To this end, we synchronized wild-type and *GAL1-DMA2* cells in G1 and released them in galactose-containing medium. At 120 and 150 min after release (end of the first cell cycle and beginning of the second cycle, respectively) cells were fixed for staining of F-actin with fluorescently labeled phalloidin. An actin ring was clearly visible at the bud neck in a small fraction of wild-type budded cells (Fig. 3d), consistent with the notion that actin is transiently recruited to the CAR shortly before constriction[22,23]. Similarly, *GAL1-DMA2* cells formed actin rings with similar efficiency at the right time. Furthermore, chains of cells appeared often with

actin rings, in agreement with lack of CAR constriction and disassembly (Fig. 3d). Consistent with normal F-actin ring assembly, the IQGAP Iqg1, which is necessary for this process[36], was recruited to the bud neck before septin splitting in all wild-type cells (*n* = 155; Supplementary Fig. 4a) and *DMA2*-over-expressing cells (*n* = 156; Supplementary Fig. 4b).

We, therefore, conclude that the cytokinesis defects of Dma2-overexpressing cells, and in particular the lack of CAR constriction, is not accounted for by inefficient actin recruitment to the division site.

**Septin destabilization drives CAR constriction in *DMA2*-overexpressing cells.** On the basis of the above results, we hypothesized that the septin collar might hamper CAR constriction. If this were the case, destabilization of septins could be sufficient to re-establish CAR constriction in mutants affecting septin ring splitting. We, therefore, introduced the *cdc12-1* temperature-sensitive mutation in *GAL1-DMA2* cells expressing Shs1-mCherry and Myo1-GFP and analyzed their behavior at semipermissive temperature (30 °C). In the majority of the cells analyzed (*n* = 47/68) Shs1 was cleared from the bud neck at the time of mitotic exit and this was immediately followed by Myo1

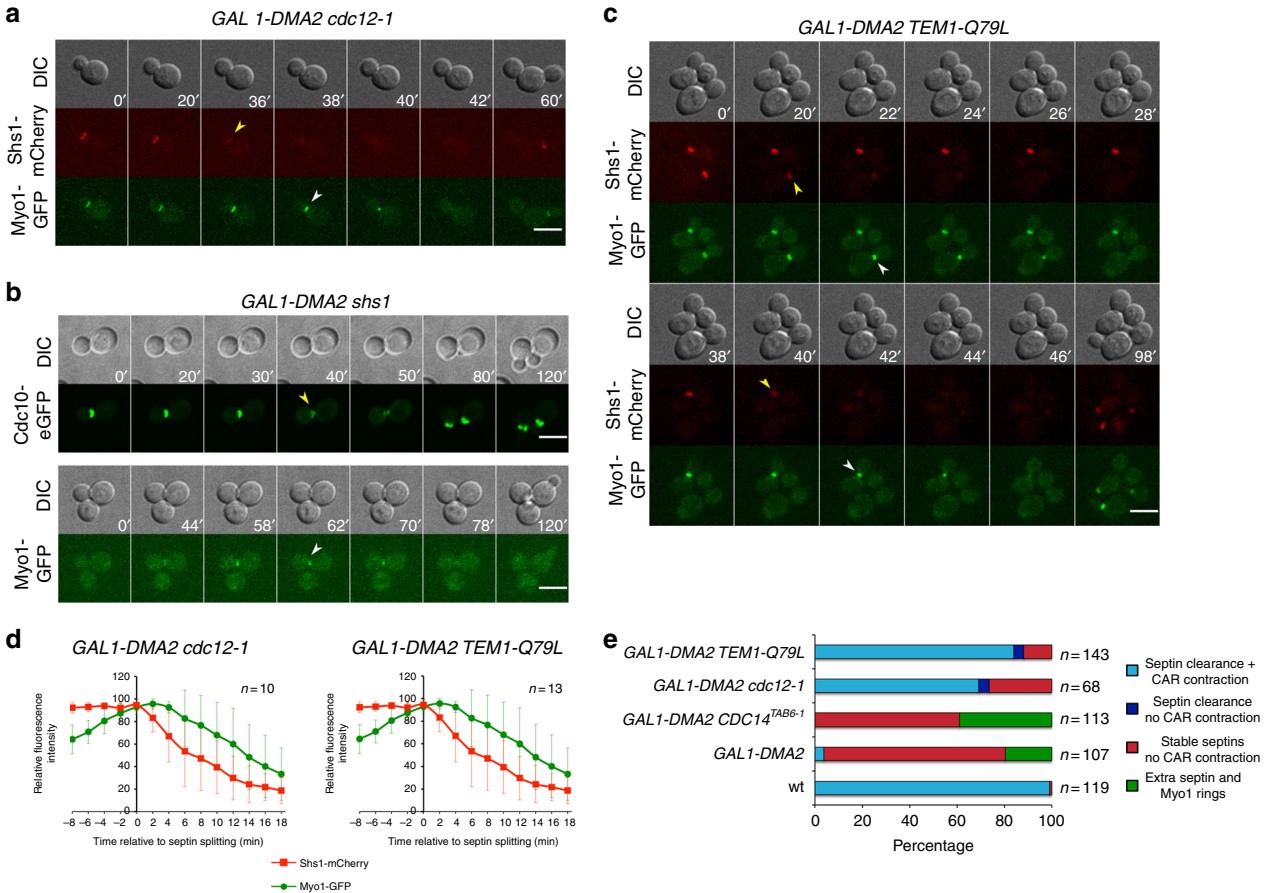

**Fig. 4** Septin destabilization triggers CAR contraction in *GAL1-DMA2* cells. **a, c** *GAL1-DMA2 BUD4* cells with the indicated genotypes and expressing Shs1-mCherry and Myo1-GFP were grown in SD-raffinose and induced for 90 min with galactose before being mounted with SD raffinose/galactose for imaging at 30 °C (every 2 min for 2 h). Arrowheads indicate disassembly of septin rings (yellow) or the onset of CAR constriction (white). DIC differential interference contrast. **b** *GAL1-DMA2 BUD4 shs1Δ* cells expressing either Cdc10-eGFP (upper panel) or Myo1-GFP (bottom panel) were treated as in **a**. Scale bar: 5 μm. **d** Quantification of fluorescence intensities associated to Shs1-mCherry and Myo1-GFP around the time of septin ring splitting (time 0) in *GAL1-DMA cdc12-1* (*n* = 10) and *GAL1-DMA2 TEM1-Q79L* cells (*n* = 13): red squares: Shs1-mCherry; green circles: Myo1-GFP. Error bars: s.d. **e** Cells with the indicated genotypes and expressing Shs1-mCherry and Myo1-GFP were induced with galactose for ~90 min and imaged every 2 min for 2 h at 30 °C in SD-raffinose/galactose. Cells were classified according to their behavior for what concerns septin ring splitting and CAR constriction

constriction (Fig. 4a, d, e), indicating that septin clearance is sufficient to drive CAR constriction upon *DMA2* overexpression. Most of the remaining cells did not undergo mitotic exit (*n* = 18/68), and therefore neither septin splitting nor CAR contraction, during the entire duration of the movie (2 h). Only a minority of cells (*n* = 3/68) underwent apparent septin clearance without CAR constriction. Deletion of the *SHS1* septin gene in *GAL1-DMA2* cells led to similar results, i.e., was sufficient for clearance of the septin collar at mitotic exit and for CAR constriction upon Dma2 overexpression (Fig. 4b).

We, therefore, conclude that septin ring splitting or clearance at the division site is an essential prerequisite for CAR constriction.

The anillin-like protein Bud4 stabilizes septin rings during splitting[8]. We, therefore, asked if deletion of *BUD4* had an impact on cytokinesis of *DMA2*-overexpressing cells. Remarkably, live cell imaging showed that 88% of *GAL1-DMA2 bud4Δ* cells (*n* = 233) underwent sudden septin disappearance in late mitosis that was shortly followed by CAR constriction (Supplementary Fig. 5a, b), further strengthening the notion that septin destabilization is sufficient to induce CAR contraction upon *DMA2* overexpression. However, in the face of an apparently normal CAR constriction, *GAL1-DMA2 cdc12-1, GAL1-DMA2 shs1Δ* and *GAL1-DMA2 bud4Δ* remained unable to accomplish full

cytokinesis, as shown by FACS analysis of DNA contents on synchronized cultures (Supplementary Figs. 5c and 6a), suggesting that late cytokinetic processes (e.g., septum formation or cell separation) might also be defective in *DMA2*-overexpressing cells.

**Dma2 prevents septin ring splitting through inhibition of MEN signaling.** Moderate overexpression of *DMA2* to levels that are well tolerated by wild-type cells was toxic for MEN mutants at permissive temperature, with *tem1–3* displaying the most dramatic synthetic phenotype (Supplementary Fig. 7 and ref. [31]). In light of these genetic interactions and given the remarkable phenotypic similarity between *GAL1-DMA2* and *tem1* or *cdc15* mutants forced to exit mitosis, we asked if Tem1 hyperactivation through the GTP-locked *TEM1-Q79L* allele[17] could promote septin ring splitting/disappearance and CAR constriction in *DMA2*-overexpressing cells. Strikingly, 84% of the *GAL1-DMA2 TEM1-Q79L* cells that we imaged for 2 h (*n* = 143) underwent septin clearance from the bud neck and CAR constriction shortly afterwards (Fig. 4c–e). Furthermore, *TEM1-Q79L* restored in most cells bud neck recruitment of Chs2, which then contracted with the CAR (Supplementary Fig. 3c, d).

These results further corroborate the idea that CAR constriction and septum formation are intimately coupled to septin ring

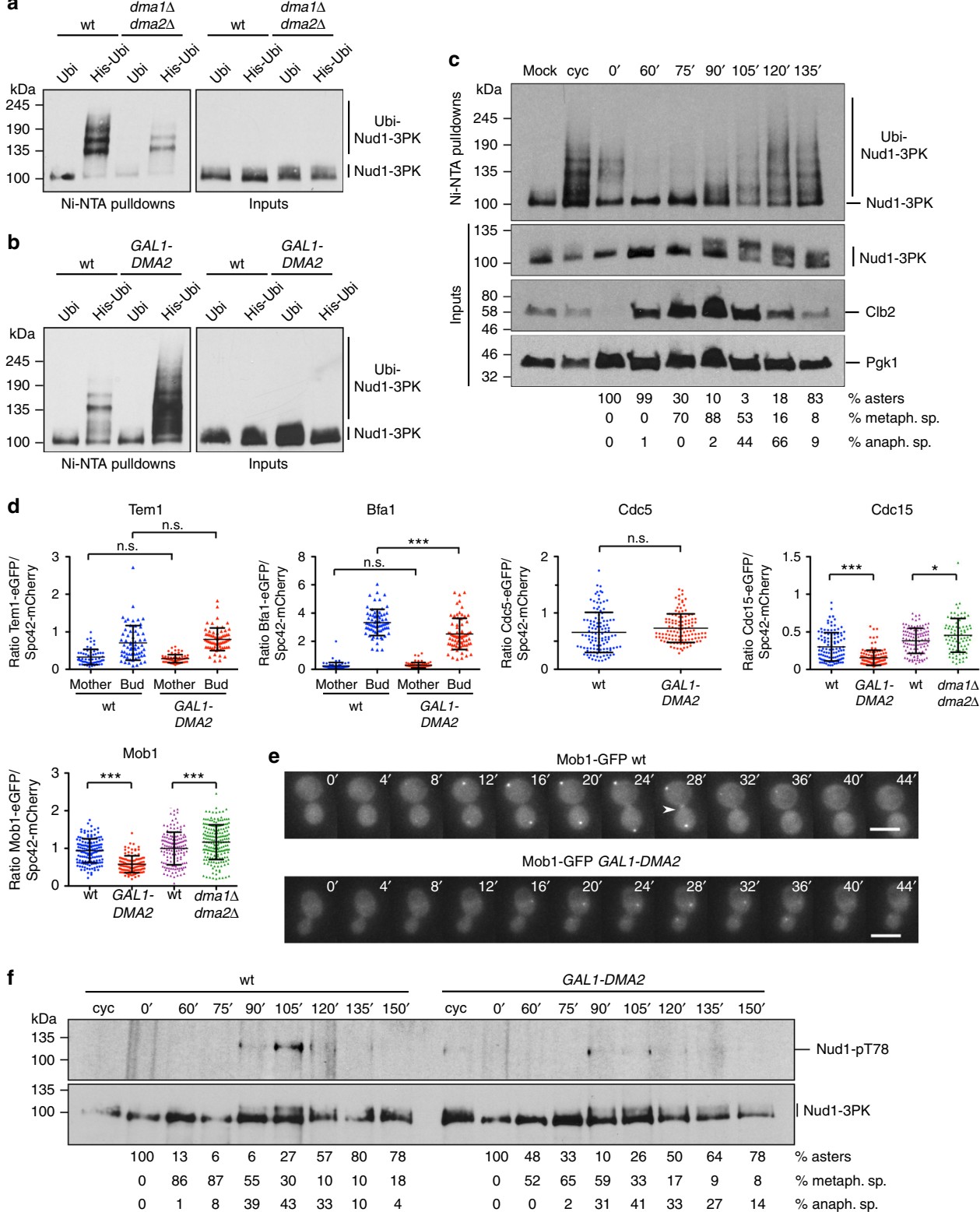

splitting. Furthermore, they suggest that high levels of Dma2 might prevent septin ring splitting along with CAR constriction and septum formation through downregulation of MEN signaling.

In spite of their apparently normal CAR constriction, *GAL1-DMA2 TEM1-Q79L* cells could not complete cell division, as shown by FACS analysis of DNA contents on synchronized cell cultures (Supplementary Fig. 6a). This result was surprising because we previously showed that deletion of the GTPase-

activating protein (GAP) Bub2 or *TEM1* overexpression could efficiently rescued the lethality and cytokinesis defects caused by an excess of Dma2[31]. Furthermore, the *TEM1-Q79L* allele rescued the lethality of *GAL1-DMA2* cells on galactose-containing media (Supplementary Fig. 6b), consistent with our previous findings. We, therefore, wondered if the presence of a wild-type copy of *BUD4* that we introduced in our W303 background to follow septin ring splitting (see Methods) could account for the

**Fig. 5** Dma1 and Dma2 promote Nud1 ubiquitination and inhibit recruitment of MEN factors to SPBs. **a–c** Ni-NTA pulldown assays were carried out using cell extracts from strains with the indicated genotypes expressing Nud1-3PK at endogenous levels and overexpressing either untagged ubiquitin or His-tagged ubiquitin from the *CUP1* promoter. Nud1 ubiquitination was revealed by western blot using an anti-PK antibody. *DMA2* was overexpressed for 30 min by addition of 1% galactose to raffinose-containing medium (**b**). For the time-course experiment in **c** cells were grown in -Trp medium (mock and cyc), arrested in G1 by alpha factor in YEPD and then released in fresh YEPD (time 0). At the indicated times cells were collected for Nud1 ubiquitination assays and tubulin immunofluorescence. Mock: Ni-NTA pulldown from cells extracts of cells expressing Nud1-3PK and untagged ubiquitin. Cyc: cycling cells. **d** Cells expressing Spc42-mCherry along with a specific MEN factor tagged with GFP/eGFP were arrested in G1 by alpha factor and then released in fresh medium at 25 °C to enrich cells in anaphase. Cells were fixed at different time points for quantifying the relative fluorescence of MEN factor vs. Spc42-mCherry in anaphase (see Methods). $n \geq 60$. Statistical significance of differences was assessed by two-tailed *t* test, assuming unequal variances (*$p <$ 0.05; **$p < 0.01$; ***$p < 0.001$; n.s.: not significant). **e** Wild-type and *GAL1-DMA2* cells expressing Mob1-GFP were imaged at 30 °C every 4 min in SD-raffinose/galactose. Fluorescent dots represent SPBs, while the arrowhead indicates in the transient appearance of Mob1 at the bud neck of wild-type cells. Scale bar: 5 μm. **f** Wild-type and *GAL1-DMA2* cells expressing Nud1-3PK were grown in YEPR, arrested in G1 with alpha factor and released in fresh YEPRG medium after 30 min induction with galactose. Cells were collected at the indicated times after release (time 0) for FACS analysis of DNA contents (Fig. S11b), in situ immunofluorescence of tubulin and for western blot detection of Nud1-pS78 and Nud1-3PK. Cyc: cycling cells

incomplete cell division of *GAL1-DMA2 TEM1-Q79L* cells. This was indeed the case: in contrast to their *BUD4* counterpart, the *TEM1-Q79L* allele in the W303 *bud4-G2459fs* background could fully rescue the cytokinesis defects of *GAL1-DMA2* cells (Supplementary Fig. 6c). The reason for this is unclear at the moment, but these data suggest that the C-terminus of Bud4 has a detrimental effect on cytokinesis under these conditions. However, in both *BUD4* and *bud4-G2459fs* backgrounds Tem1 hyperactivation was sufficient to destabilize septins in late telophase in cells overexpressing *DMA2*, thereby allowing at least some cytokinetic events and cell proliferation.

**Dma2 promotes ubiquitination of the MEN scaffold at SPBs Nud1**. The septins Cdc11 and Shs1 were previously shown to be ubiquitinated by Dma1 and Dma2[37], which could underlie the mechanism by which Dma2 inhibits septin ring splitting. We re-investigated this issue using Ni-NTA pulldowns of ubiquitinated proteins from cells overexpressing untagged or His-tagged ubiquitin, followed by western blot to detect Cdc11-HA or Shs1-HA expressed at endogenous levels from their genomic loci. Unexpectedly, deletion of both *DMA1* and *DMA2* in our genetic background did not reduce the ubiquitination levels of either Cdc11 or Shs1, but conversely increased them (Supplementary Fig. 8a, b). Additionally, although *DMA2* overexpression induced hyper-ubiquitination of both Cdc11 and Shs1 (Supplementary Fig. 8c, d), in agreement with previous data[37], this was not suppressed by the *TEM1-Q79L* allele that allows septin clearance in *DMA2*-overexpressing cells (Supplementary Fig. 8e), suggesting that other targets might be instrumental for Dma1/2-dependent inhibition of septin ring splitting.

We considered that Tem1 could be a good candidate. Using the same experimental setup that we used for septins, we could clearly detect Tem1 ubiquitination in yeast extracts, consistent with previous data[38]. However, Tem1 ubiquitination was not affected by either *DMA1/2* deletion or *DMA2* overexpression (Supplementary Fig. 8f, g), suggesting that Tem1 is not ubiquitinated by Dma1/2.

The constitutive SPB component Nud1 is required for MEN signaling and mitotic exit by recruiting Tem1, Cdc15, and Mob1-Dbf2/20 in a hierarchical manner, thereby leading to Cdc14 release from the nucleolus[15,16,18,19]. Since Dma1, like its counterpart in Schizosaccharomyces *pombe*, is present at SPBs[39,40] we reasoned that Nud1 could be a likely target of Dma1/2. Furthermore, a small fraction of 3HA-tagged Dma2 co-immunoprecipitated with 3Flag-tagged Nud1 in anaphase (Supplementary Fig. 9), suggesting that the two proteins physically interact in a cell cycle-regulated fashion. Strikingly, using Ni-NTA pulldown assays as above we found that

ubiquitination of Nud1 was markedly affected by deletion of both *DMA1* and *DMA2* (Fig. 5a), while it was conspicuously upregulated by a short induction of *GAL1-DMA2* (Fig. 5b). To get further insights into its physiological significance, we analyzed Nud1 ubiquitination throughout the cell cycle after G1 arrest and release of cells for different times. Interestingly, Nud1 ubiquitination was low in S, G2, and M but markedly induced from mitotic exit to G1 (Fig. 5c), suggesting that Nud1 ubiquitination by Dma2 might silence MEN signaling. Upon *DMA2* overexpression Nud1 ubiquitination was enhanced most markedly between late mitosis and G1 (Supplementary Fig. 10a). Furthermore, it could be steered upon *GAL1-DMA2* induction in cells arrested in mitosis by nocodazole treatment, but not in cells arrested in S phase by hydroxyurea (Supplementary Fig. 10b). Altogether, these data suggest that Nud1 might be a direct ubiquitination target of Dma1/2 in late M and G1 phase.

We then investigated the consequences of *DMA2* overexpression on the SPB recruitment of MEN factors specifically in anaphase, when the presence of MEN factors at SPBs reaches its peak. To this end, we calculated the ratio between the fluorescence intensity of MEN proteins tagged with GFP and that of the constitutive SPB component Spc42 tagged with mCherry. Additionally, since Tem1 and its GAP Bub2-Bfa1 localize asymmetrically at SPBs and are more concentrated on the bud-directed SPB[17,41,42], we further distinguished the mother- from the bud-confined SPB. Recruitment of Tem1 and the polo kinase Cdc5, which promotes Tem1 activation by inhibiting the Bub2-Bfa1 GAP[14], was unaffected by *DMA2* overexpression. Conversely, localization of Bub2-Bfa1, Cdc15, and Mob1 at SPBs was inhibited under the same conditions (Fig. 5d and Supplementary Fig. 11a). Furthermore, SPB recruitment of Cdc15 and Mob1 was mildly but significantly stimulated upon deletion of *DMA1* and *DMA2* (Fig. 5d), suggesting that Nud1 ubiquitination by Dma1/2 antagonizes MEN signaling by attenuating its scaffolding activity toward MEN. Interestingly, localization of Mob1-GFP to the bud neck at cytokinesis[27] was also impaired in *GAL1-DMA2* cells, perhaps as a consequence of its reduced recruitment to SPBs. Indeed, while wild-type cells transiently showed Mob1-GFP at the bud neck in 43/70 cells during the time frame occurring between its appearance and disappearance at SPBs, only 5/60 *GAL1-DMA2* cells did so (Fig. 5e).

As an additional readout of MEN activity at SPBs, we monitored the Cdc15-dependent Nud1 phosphorylation on Ser78[16] throughout the cell cycle of wild-type and *GAL1-DMA2* cells. Remarkably, while Nud1 S78 phosphorylation peaked in late mitosis in wild-type cells, consistent with previous data[16], it was largely suppressed upon *DMA2* overexpression (Fig. 5f and Supplementary Fig. 11b). Furthermore, total Nud1

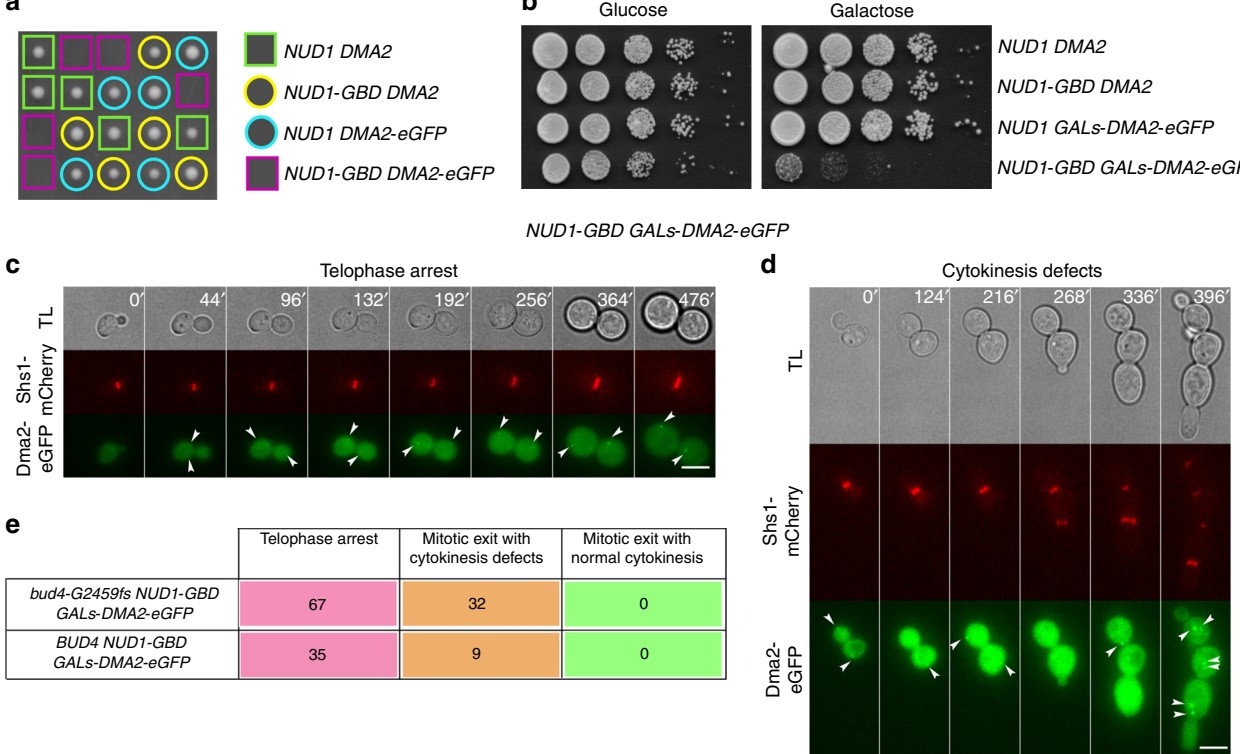

**Fig. 6** Constitutive association between Dma2 and Nud1 prevents mitotic exit and cytokinesis. **a** Meiotic segregants obtained after sporulation of diploid cells generated by crossing *NUD1-GBD* with *DMA2-eGFP* haploid cells. Genotypes were confirmed by PCR. **b** Serial dilutions of cells with the indicated genotypes were spotted on YEPD and YEPG plates and incubated at 30 °C. **c–e** *NUD1-GBD GALs-DMA2-eGFP* cells, either *BUD4* or *bud4-G2459fs*, expressing Shs1-mCherry and grown in SD-raffinose were induced for 90 min with galactose and then imaged in SD-raffinose/galactose at 30 °C every 4 min. **c** Telophase arrest; **d** cytokinesis defects; **e** number of cells showing the indicated phenotypes in the movies. Arrowheads indicate Dma2-eGFP at SPBs. TL transmitted light. Scale bar: 5 μm

phosphorylation, which requires Cdc15 and Cdc5[16,43], was maximal at mitotic exit (i.e., when the levels of Cdc5 started decreasing) in wild-type cells, as judged by its reduced electrophoretic mobility on sodium dodecylsulfate polyacrylamide gel electrophoresis (SDS-PAGE) ($t = 105$ min, Supplementary Fig. 11c), but impaired upon *DMA2*-overexpression. Conversely, phosphorylation of the SPB component Spc72, which depends on Cdc5[43], was unaffected (Supplementary Fig. 11c). We, therefore, conclude that Cdc15 kinase activity is downregulated at SPBs upon Nud1 ubiquitination by Dma1/2, while the Cdc5 kinase remains active under the same conditions, consistent with our previous conclusions[31].

To further strengthen the notion that Dma2 acts as a MEN inhibitor at SPBs through Nud1 ubiquitination, we forced the constitutive association between Dma2 and Nud1 by tagging the latter with a GFP-nanotrap (GFP-binding domain or GBD[44],) and expressing in the same cells Dma2-eGFP. Tetrad analysis after genetic crosses and sporulation revealed that the combination *NUD1-GBD DMA2-eGFP* was lethal (Fig. 6a). To analyze the phenotype of these cells, we generated a conditional mutant by placing *DMA2-eGFP* under the control of the attenuated galactose-inducible *GALs* promoter[45]. The resulting *GALs-DMA2-eGFP* construct was perfectly tolerated by otherwise wild-type cells, while it was toxic for *NUD1-GBD* cells in galactose-containing medium (Fig. 6b). Live cell imaging of *NUD1-GBD GALs-DMA2-eGFP* cells expressing Shs1-mCherry and dividing in the presence of galactose showed that the majority of cells arrested in late mitosis as large budded cells with unsplit septin rings at the bud neck (Fig. 6c, e), consistent with MEN inhibition. Another fraction of cells could eventually exit mitosis,

but displayed severe cytokinesis defects (Fig. 6d, e). During this analysis, we noted that the presence of full length *BUD4* was deleterious for *NUD1-GBD GALs-DMA2-eGFP* cells already in raffinose-containing medium (i.e., noninduced conditions), causing them to prematurely die and often stop dividing, while *NUD1-GBD GALs-DMA2-eGFP* cells carrying the truncated *bud4-G2459fs* allele of W303 (see Methods) were healthy in the same conditions and stopped dividing only after galactose induction, suggesting that the C-terminus of Bud4 might somehow compromise MEN signaling under these sensitized conditions.

Altogether, our data clearly indicate that Dma2 is a powerful inhibitor of MEN signaling at SPBs.

**Cdc14 recruitment to SPBs promotes septin clearance at the bud neck.** Since *DMA2* overexpression weakens SPB localization of several MEN factors, which in turn are critical for the transient recruitment of the Cdc14 phosphatase to the bud-directed SPB in anaphase[46,47], we asked if the latter was similarly impaired in *GAL1-DMA2* cells. Live cell imaging of cells expressing Cdc14-eGFP at endogenous levels confirmed that Cdc14 appears at the bud-directed SPB in a large fraction of wild-type cells during anaphase (86%, $n = 56$; Fig. 7a). Strikingly, upon *DMA2* over-expression the fraction of anaphase cells displaying SPB-localized Cdc14 dropped to 24,6% ($n = 134$). Furthermore, in the few cells where Cdc14-eGFP lighted up at the SPB its signal was more transient and weak than in wild-type cells (Fig. 7a), suggesting that lower levels of Cdc14 are recruited to the bud-directed SPB in *GAL1-DMA2* cells. We further note that the release of Cdc14

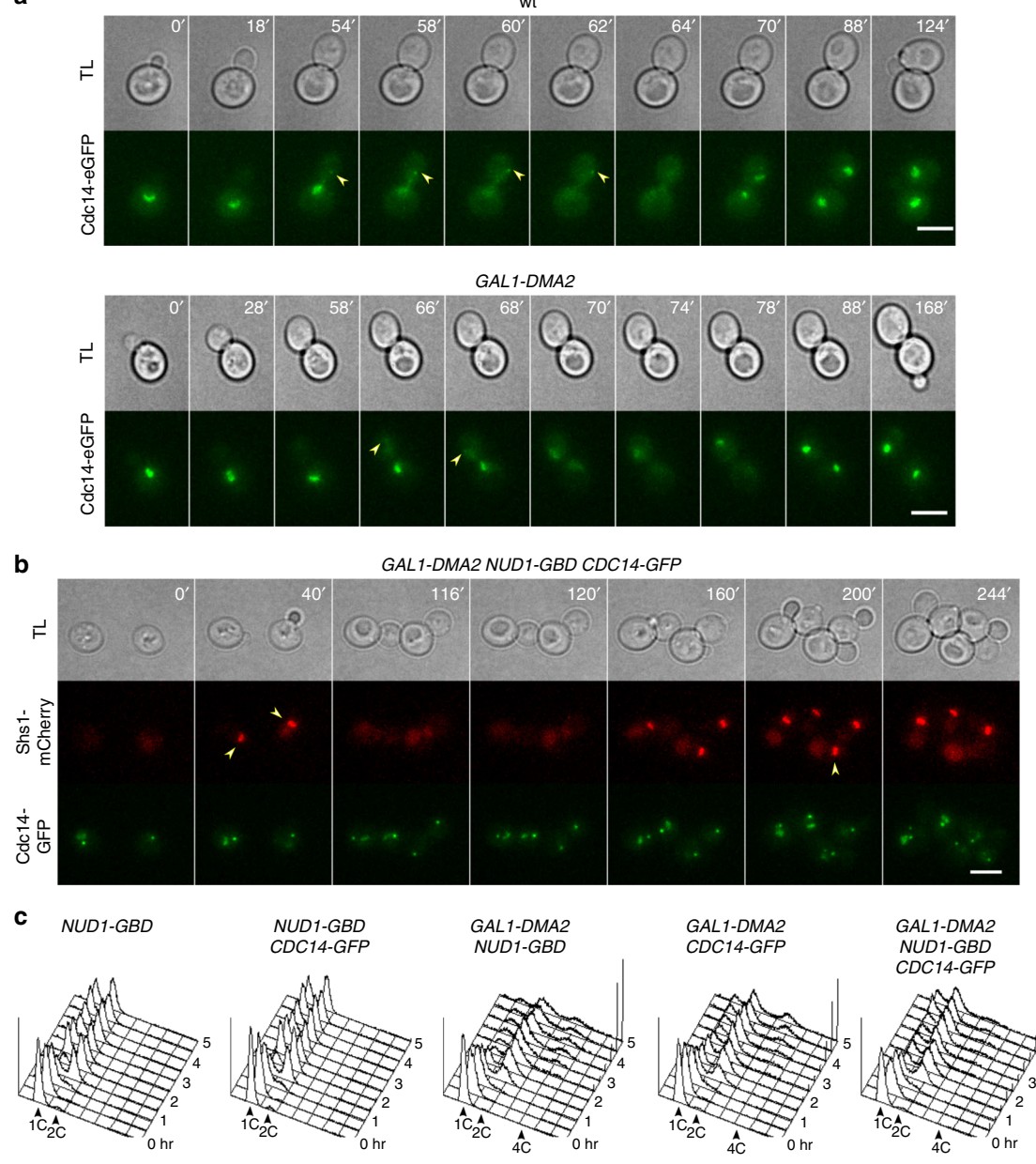

**Fig. 7** Constitutive recruitment of Cdc14 to SPBs suppresses the cytokinetic defects of *GAL1-DMA2* cells. **a** *BUD4* and *GAL1-DMA2 BUD4* cells expressing Cdc14-eGFP at endogenous levels were imaged by time lapse fluorescence microscopy at 30 °C every 2 min. Scale bar: 5 μm. **b** *GAL1-DMA2 BUD4* cells expressing Nud1-GBD at endogenous levels and Cdc14-GFP from a centromeric plasmid were imaged at 30 °C every 4 min in selective medium (−His containing raffinose and galactose) after being induced for ~90 min with galactose. TL transmitted light. Arrowheads indicate the appearance of Cdc14-eGFP at the bud-directed SPB in anaphase. Scale bar: 5 μm. **c** Cells with the indicated genotypes (all *BUD4*) were grown in selective medium (−His containing raffinose) at 25 °C, arrested in G1 with alpha factor, induced with galactose 30 min before the release and finally released in YEPRG at 30 °C. At the indicated times cells were collected for FACS analysis of DNA contents. FACS data were plotted after gating out the debris as illustrated in Supplementary Fig. 12

from the nucleolus was also somewhat impaired in the *GAL1-DMA2* mutant, with many cells showing only partial or ephemeral release, consistent with the idea that high levels of Dma2 interfere with MEN signaling.

If lack of septin ring splitting in *GAL1-DMA2* cells were due to insufficient levels of Cdc14 at SPBs, we might expect to restore efficient splitting and cytokinesis by artificially recruiting Cdc14 to the SPB. To force constitutive anchoring of Cdc14 to SPBs, we expressed in *NUD1-GBD* cells an extra copy of the *CDC14* gene fused to GFP. In fact, given the plethora of functions that Cdc14 plays in several processes, we reasoned that constitutive

anchorage of all Cdc14 at SPBs would be lethal. This strategy robustly recruited Cdc14 to mother and daughter SPB throughout the cell cycle (Fig. 7b and Supplementary Movie 1). Remarkably, this was sufficient to cause septin disappearance from the bud neck and to promote cytokinesis in 97% of *DMA2*-overexpressing cells (*n* = 337; Fig. 7b and Supplementary Movie 1). This was confirmed on synchronized cell populations that were released from a G1 arrest in the presence of galactose to induce *GAL1-DMA2* overexpression (Fig. 7c): while *GAL1-DMA2 NUD1-GBD* cells underwent cytokinesis defects that caused the accumulation of cells with DNA contents ≥2C in the second cell cycle, *GAL1-*

*DMA2 NUD1-GBD CDC14-GFP* cells could efficiently divide and transiently piled up in the next G1 with 1C DNA content. Expression of *CDC14-GFP* in *GAL1-DMA2* cells lacking Nud1-GBD increased only modestly their ability to undergo cytokinesis (Fig. 7c), indicating that Cdc14 recruitment to the SPB, rather than increased levels of *CDC14* expression, is responsible for restoring clearance of the septin ring from the division site.

## Discussion

Although we know since long that in budding yeast CAR constriction takes place between split septin rings, the role and regulation of septin ring splitting has remained mysterious, mainly due to the lack of mutants specifically affecting this process. Previous evidence that Tem1 depletion prevents both septin ring splitting and CAR constriction in cells that are forced to release Cdc14 from the nucleolus[7] did not rule out the possibility that some MEN components are involved in both processes independently of their role in mitotic exit. Our data show that during an unperturbed cell cycle septin ring splitting precedes temporally CAR constriction and no physical connections can be detected between septins and CAR by SIM microscopy during cytokinesis. Furthermore, our results firmly establish that septin displacement from the division site is an absolute requirement for subsequent CAR constriction and cytokinesis. Indeed, mutants affecting septin splitting not only invariably fail to undergo CAR constriction, but septin destabilization (through septin mutant alleles or deletion of the anillin Bud4) also causes disappearance of septins from the bud neck during mitotic exit and is sufficient to promote CAR constriction. Thus, while being necessary for recruitment of CAR components to the bud neck, eviction of the septin collar from the division site is likewise essential for cytokinesis to take place. This provides an intrinsic safe-lock mechanism that ensures the correct temporal order of cytokinetic events.

This mechanism may be conserved in other organisms. In fission yeast, where a septin ring at the medial site is involved in septation but dispensable for CAR assembly[48,49], the septin ring also splits in two before cytokinesis, suggesting that septin ring splitting could facilitate CAR constriction. Conversely in *Drosophila*, where septins bundle actin filaments for CAR assembly, septins are integral part of the CAR and constrict with it[4,50].

How exactly the septin ring restrains CAR constriction in yeast is a crucial question to be addressed in the future. We show here that lack of septin ring splitting also restrains recruitment to the bud neck of the chitin synthase Chs2, whose synthesis of the primary septum is coupled to CAR constriction[1,2]. However, CAR contraction initiates in the absence of Chs2, suggesting that other factors must be invoked to explain the inhibitory effects of septins on this process. One possibility is that the septin collar acts as a physical constraint by either preventing proper contacts between CAR and plasma membrane or harnessing the CAR in a nonconstrictable arrangement.

The best understood function of MEN is to promote activation of the Cdc14 phosphatase, thereby leading to inactivation of mitotic CDKs (reviewed in ref. [20]). Since persistent CDK activity inhibits cytokinesis in many organisms, it is not surprising that MEN mutants are unable to bring about cytokinetic events, including septin ring splitting and CAR contraction. An important issue, however, is whether MEN factors play a direct role in cytokinesis beyond Cdc14-mediated CDK inhibition. Indeed, several MEN factors relocalize to the bud neck in late anaphase. MEN mutants that are allowed to exit mitosis in restrictive conditions through forced nucleolar release of Cdc14 (e.g., by *NET1* deletion or the dominant *CDC14^{TAB6-1}* allele) or inhibition of mitotic CDKs (e.g., by overexpression of the CDK inhibitor

Sic1) have been used to address this issue. This strategy has allowed establishing a key role for the Dbf2 kinase in septum formation through direct phosphorylation of the chitin synthase Chs2 and its regulatory complex Hof1-Inn1-Cyk3[30,51]. Furthermore, it has implicated the Tem1 GTPase in septin ring splitting and CAR constriction[7]. We have confirmed and extended this result, by showing that the Cdc15 kinase and Mob1 are also required for these processes downstream of Tem1. Surprisingly, the Dbf2/Dbf20 kinase seems to be dispensable for septin ring splitting, as shown by the behavior of mutants where Dbf2 can be either heat-inactivated or depleted through an auxin-inducible degron in cells lacking Dbf20. Although we cannot rule out that residual amounts of Dbf2 are sufficient for septin splitting in our experimental set-up, our results are consistent with published data[29,30]. Thus, Mob1 could promote septin ring splitting by associating to kinases other than Dbf2/Dbf20, or by directly enforcing Cdc14 recruitment to the SPB.

Our study has revealed a key role for SPB-localized Cdc14 in septin clearance from the bud neck that is independent of its mitotic exit function. The use of *DMA2*-overexpressing cells that are specifically defective in septin ring splitting and cytokinesis (see below), while undergoing mitotic exit[31], has been instrumental to this discovery. Although Cdc14 might be essential for mitotic exit only in budding yeast, its requirement for cytokinesis seems conserved[52,53]. Potential and established Cdc14 cytokinetic targets have been identified[54–58]. Although the critical substrates of budding yeast Cdc14 in septin ring splitting remain to be determined, potential candidates include Bud4 and the septin-associated kinase Gin4[54,59]. Additionally, Cdc14 could promote this process also by potentiating MEN signaling through positive feedback controls[60,61]. The identification of MEN targets in septin ring splitting will be key to further dissect the mechanistic details underlying this process.

Our previous genetic data suggested that Dma1/2 might negatively regulate MEN signaling through an unknown mechanism[31]. Several new observations described here are consistent with this hypothesis: (i) MEN mutants are hypersensitive to moderate Dma2 overexpression; (ii) the cytokinetic defects of Dma2-overexpressing cells are remarkably similar to those of MEN mutants undergoing mitotic exit through the *CDC14^{TAB6-1}* allele; (iii) constitutive recruitment of Dma2 to SPBs delays/ blocks mitotic exit; and (iv) the cytokinetic defects of *GAL1-DMA2* cells are bypassed by a GTP-locked variant of Tem1 (Tem1-Q79L[17]). Altogether, these observations indicate that *GAL1-DMA2* is a hypomorphic MEN mutant. Consistently, upon *DMA2* overexpression Cdc14 nuclear release is somewhat impaired, and degradation of some APC^{Cdh1} targets, such as Cdc5, but not Clb2[31], is delayed. We now show that Dma1/2 promote ubiquitination of the SPB component Nud1, thereby weakening its ability to recruit and activate MEN factors, such as Cdc15, Mob1, Bub2-Bfa1, and Cdc14. We further show that Nud1 ubiquitination takes place in late mitosis, presumably after cytokinesis, and is rapidly induced by Dma2 overexpression preferentially in mitosis and G1 phase, suggesting that Dma2 might directly ubiquitinate Nud1. The finding that Dma1 (and presumably Dma2) localize at SPBs in late mitosis[40] is consistent with this conclusion. We envisage that the physiological role of Nud1 ubiquitination by Dma1/2 during the normal cell cycle is to turn down MEN signaling at SPBs after cytokinesis (see Fig. 8). Indeed, Cdc14 inactivation and re-entrapment in the nucleolus is likely an important step for the subsequent cell cycle, as persistent release of Cdc14 from the nucleolus interferes with DNA replication[62]. Since deletion of *DMA1* and *DMA2* is well tolerated by yeast cells, redundant mechanisms obviously participate to timely MEN silencing after mitotic exit and cytokinesis in unperturbed conditions. One of them is degradation of the polo kinase Cdc5

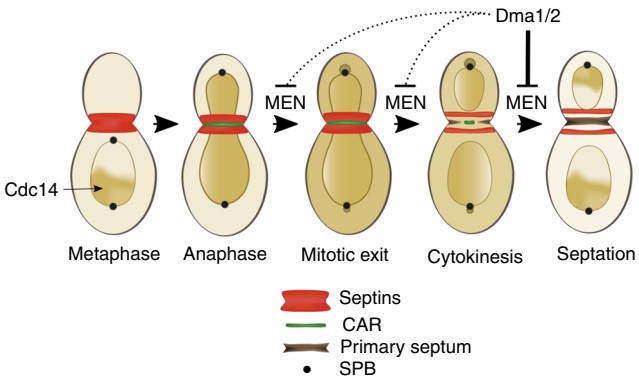

**Fig. 8** Model. In metaphase Cdc14 is trapped in the nucleolus, while at the onset of anaphase it diffuses into the nucleus. At this stage, as well as during previous cell cycle phases (not depicted), septins form a scaffolding collar at the bud neck that recruits cytokinesis factors, including CAR components. In late anaphase/telophase the mitotic exit network (MEN) promotes Cdc14 full release into the cytoplasm, which allows its recruitment also to SPBs (preferentially the bud-directed SPB) and brings about mitotic exit. MEN activity and SPB-localized Cdc14 then drive septin ring splitting, which is in turn an essential prerequisite for CAR contraction and septum formation. At the end of cytokinesis the Dma1 and Dma2 ubiquitin ligases contribute to shut-off MEN signaling at SPBs via ubiquitination of their scaffold Nud1. However, when overexpressed or constitutively bound to Nud1, Dma2 acts as a MEN potent inhibitor and interferes with MEN-dependent functions, mainly for what concerns cytokinesis and, to a lesser extent, mitotic exit. See text for details

by APC[Cdh1], which in turn is activated by Cdc14 itself[63]. Another is reactivation of the GAP Bub2-Bfa1 at SPBs by Cdc14-mediated dephosphorylation[46]. Thus, Cdc14 sets the stage for its own inhibition and return to the nucleolus. In the future, it will be interesting to investigate if Dma-dependent Nud1 ubiquitination is also modulated by Cdc14. The finding that Dma2 is a potential Cdc14 substrate[54] makes this hypothesis very appealing.

Though dispensable during the unperturbed cell cycle, the role of Dma1/2 in MEN inhibition becomes crucial upon spindle mispositioning, when these E3 ligases participate to the checkpoint that couples cytokinesis to proper chromosome segregation[31,32]. Other adverse conditions negatively impact on MEN activation. For instance, failure to properly segregate mitochondria during mitosis leads to MEN inhibition[64]. Whether Dma1/2 plays any role in this process remains to be addressed. However, it is tempting to speculate that Nud1 ubiquitination by Dma1/2 could be critical for coupling cytokinesis to proper segregation of organelles as well as of chromosomes, thereby ensuring equal ploidy and metabolic capacity to daughter cells.

Several lines of evidence have established the importance of MEN signaling at SPBs in the regulation of mitotic exit[15–19]. Our data clearly indicate that MEN signaling at SPBs is also crucial for septin ring splitting (see Fig. 8). Not only lack of septin splitting correlates with decreased levels of MEN factors at SPBs in Dma2-overexpressing cells, but constitutive recruitment of Cdc14 to SPBs in these cells is sufficient to restore septin clearance and cytokinesis. It is worth noting, however, that under these conditions septins suddenly disappear from the bud neck, rather than splitting, suggesting that the activity of septin stabilizers during splitting, like Bud4, might be perturbed.

A key role for SPBs/centrosomes during cytokinesis is clearly emerging in several organisms. For instance, laser ablation of both SPBs in fission yeast leads to cytokinesis failure[65]. The fission yeast counterpart of Nud1, Cdc11, promotes SIN signaling and cytokinesis by scaffolding SIN components at the SPBs

together with Sid4[66,67]. The human protein centriolin, which shares homology with *S.c.* Nud1 and *S.p.* Cdc11, resides at centrioles and is required for abscission, the latest cytokinetic step[68]. Likely, recruitment of cytokinesis-promoting factors to the SPB/centrosome is an imperative, yet intermediate stop in their journey toward the division site, in order to get fully active before proceeding to their final destination and targets. Consistently, Nud1 ubiquitination by Dma1/2 not only lowers the levels of Mob1 at SPBs, but also prevents its translocation to the bud neck in late anaphase. Considering that Nud1 is required for Mob1 localization at the bud neck[27], we hypothesize that its inability to reach the division site in Dma2-overexpressing cells is a mere consequence of its lousy activation at SPBs.

It is worth noting that the E3 ligase Dma1 in *S. pombe* negatively controls SIN signaling by ubiquitinating the SPB component Sid4 (related to budding yeast Cnm67), which in turn recruits the Nud1-related protein Cdc11 and downstream SIN factors[66,67,69]. This leads to a decrease in polo kinase levels at SPBs, thereby preventing cytokinesis upon spindle depolymerization[39,70]. We find that *DMA2* overexpression in budding yeast does not interfere with recruitment of the polo kinase Cdc5 to SPBs. However, it is remarkable how the two yeasts, which are evolutionary as distant from one another as each of them is distant from humans, have adopted similar, though distinct, strategies to silence MEN/SIN. Thus, an exciting possibility is that other eukaryotes might have evolved related mechanisms to prevent cytokinesis under adverse conditions in order to preserve genome stability.

## Methods

**Strains and growth conditions**. All yeast strains (Table S1) are congenic to or at least four times backcrossed to W303 (*ade2-1*, *trp1-1*, *leu2-3,112*, *his3-11*, and *15 ura3*). W303 bears a single nucleotide deletion in the *BUD4* gene (*bud4-G2459fs*) that results in a premature stop codon. The *bud4-G2459fs* gene produces a truncated protein of 838 aminoacids that lacks 609 aminoacids and carries 18 non-natural aminoacids at C-terminus (https://www.yeastgenome.org). All strains used for time-lapse video microscopy to look at septin ring splitting/disappearance have been corrected to carry full length *BUD4* unless specified. It should be noted that *DMA2* overexpression prevents septin ring splitting in both the original *bud4-G2459fs*[32] and the corrected *BUD4* background.

Yeast cultures were grown at 25−30 °C, unless otherwise specified, in either SD supplemented with the appropriate nutrients or YEP (1% yeast extract, 2% bactopeptone, 50 mg/l adenine) medium. Raffinose was supplemented to 2% (SD-raffinose and YEPR), glucose to 2% (SD-glucose and YEPD), and galactose to 1% (SD-raffinose/galactose and YEPRG). Cells were synchronized in G1 by alpha factor (4 μg/ml) in YEP medium containing the appropriate sugar at 23−25 °C. G1 arrest was monitored under a transmitted light microscope and cells were released in fresh medium (typically after 120–135 min of alpha factor treatment) after being collected by centrifugation at 2000*g* and washed with YEP containing the appropriate sugar. IAA (3-indoleacetic acid) was dissolved in ethanol as 1000× stock solutions and used at a final concentration of 0.1–0.25 mM.

Generation and integration in the genome of the *GAL1-DMA2* construct has been described[31]. The *CDC14-GFP* plasmid was a generous gift from A. Fatica. One-step tagging techniques were used to generate 3HA-, 3PK-, 3Flag-, GFP, eGFP-, mCherry-, 1XminiAID-, 3XminiAID-, GBD-tagged proteins at the C terminus. A flexible linker of six glycines was introduced between the last aminoacid and the tag when tagging Cdc10 and Cdc14 with eGFP and for tagging Nud1 with 3Flag. *MYO1-GFP* was a generous gift of J. Pringle; *SPC42-mCherry* of E. Schwob; *GFP-MOB1* of F. Luca; *GFP-CDC12* of Y. Barral; *CDC11-HA* and *SHS1-HA* of E. Johnson; *CHS2-GFP* of E. Conibear. *IQG1-GFP* was derived from strain BY25825 of the YGRC that was provided by the NBRP of the MEXT, Japan.

**Primers used in this study for gene tagging**. Sequences in bold anneal to the tag-bearing cassette

**SP223** (tagging *DMA2* with *3HA::K.l.URA3*; fwd)
GAAGGTGATCAACTGGTGGATCAACTTAGCGTCTTAATGGAAACTTCAA
AGGATGTTGATAGCCATCCT**TCCGGTTCTGCTGCTAG**

**SP224** (tagging *DMA2* with *3HA::K.l.URA3*; rev)
ATATTAAGGTACGAGATGTGGAGTTCGGTGGTTTTTCTTTATTTTTCA
AACTGTGTATTTTCTTTGACC**CCTCGAGGCCAGAAGAC**

**SP247** (tagging *CDC15* with *GFP::kanMX*; fwd)
CAAAGATAAAAGTGACGGCTTTTCCGTCCCCATTACAACATTTCAA
ACA**CGGATCCCCGGGTTAATT**

SP248 (tagging *CDC15* with *GFP::kanMX*; rev)
ATGCTGTATTATTTCTCTATATATGTATGTATGCACATGCAATTCCTA
C**GAATTCGAGCTCGTTTAAAC**

SP789 (tagging *BFA1* with *eGFP::kanMX*; fwd)
AAATCCTATATGTATGAAATCAGGAACATGGTAATCAATTCGACAAA
AGAT**GGTGACGGTGCTGGGTTTA**

SP790 (tagging *BFA1* with *eGFP::kanMX*; rev)
TGAATGTACTCAAGATAACGGTAAAGAAACAGTTATAAGAAGGCTAA
AGGG**TCGATGAATTCG**

MP218 (tagging *SHS1* with *mCherry::hphMX*; fwd)
AAAAAAAATGACACGTATACTGATTTAGCCTCTATTGCATCGGGTAG
AGA**TGGTCGACGGATCCCCGGG**

MP219 (tagging *SHS1* with *mCherry::hphMX*; rev)
CATTTATTTATTTATTTATTTGCTCAGCTTTGGATTTTGTACAGATAC
AAC**ATCGATGAATTCGAGCTCG**

MP247 (tagging *TEM1* with *3HA::K.l.URA3*; fwd)
AAGAGCCATAATATCAGGAAGCCCTCCTCGTCGCCCTCATCTAAGGC
ACCATCGCCGGGCGTTAATACAT**CCGGTTCTGCTGCTAG**

MP248 (tagging *TEM1* with *3HA::K.l.URA3*; rev)
TATTGTGTAGCTTGATTTAAAATATGCTAACGCCAACTCTCACATGGT
AGCAGGCGGGTATAGTTGTTT**CCTCGAGCCAGAAGAC**

MP547 (tagging *CDC5* with *eGFP::kanMX*; fwd)
CTTTGATAAAGGAAGGTTTGAAGCAGAAGTCCACAATTGTTACCGT
AGAT**GGTGACGGTGCTGGTTTA**

MP548 (tagging *CDC5* with *eGFP::kanMX*; rev)
CAATGGACTGGTAATTTCGTATTCGTATTTCTTTCTACTTTAATATTG
GTT**CGATGAATTCGAGCTCG**

MP631 (tagging *CDC10* with *6Gly-eGFP::kanMX*; fwd)
AATGCGAATAGTCGTTCCTCAGCTCATATGTCTAGCAACGCCATTCA
ACGT**GGGGGAGGCGGGGGTGGAGGTGACGGTGCTGGTTTA**

MP632 (tagging *CDC10* with *6Gly-eGFP::kanMX*; rev)
TGAGAATTCTTAATAACATAAGATATATAATCACCACCATTCTTATGA
GATT**CGATGAATTCGAGCTCG**

MP664 (tagging *CDC14* with *6Gly-eGFP::kanMX*; fwd)
CTACAAGCGCCGCCGGTGGTATAAGAAAAATAAGTGGCTCCATCAAG
AAA**GGGGGAGGCGGGGGTGGAGGTGACGGTGCTGGTTTA**

MP665 (tagging *CDC14* with *6Gly-eGFP::kanMX*; rev)
TAGTAAGTTTTTTTATTATATGATATATATATATATAAAAATGAAATA
AA**TCGATGAATTCGAGCTCG**

MP757 (tagging *NUD1* with *3PK::K.l.HIS3*; fwd)
CTGGCGACCCTCTGGTTAGATGACACTCCTGCCCCAACTGCCACGAA
TCTG**TCCGGTTCTGCTGCTAG**

MP758 (tagging *NUD1* with *3PK::K.l.HIS3*; rev)
ATTTACTAATTACATACATTTTTAGTACTGCGTACGGGTATAGTTATG
GGG**CCTCGAGGCCAGAAGAC**

MP813 (tagging *NUD1* with *GBD::kanMX*; fwd)
CTGGCGACCCTCTGGTTAGATGACACTCCTGCCCCAACTGCCACGAA
TCTG**CGTACGCTGCAGGTCGAC**

MP814 (tagging *NUD1* with *GBD::kanMX*; rev)
ATTTACTAATTACATACATTTTTAGTACTGCGTACGGGTATAGTTATG
GGG**ATCGATGAATTCGAGCTCG**

MP853 (tagging *DBF2* with *1-3XminiAID::kanMX*; fwd)
GGCATCTTATTCAACGGACTGGAACACTCAGACCCCTTTTCAACCTTT
TAC**CGTACGCTGCAGGTCGAC**

MP854 (tagging *DBF2* with *1-3XminiAID::kanMX*; rev)
GTTAAAGCTAATTATATCGCGGCGAATGCAAGACAAGAATTCATTTT
TACG**ATCGATGAATTCGAGCTCG**

MP1004 (tagging *NUD1* with *6Gly-3Flag::kanMX*; fwd)
CTGGCGACCCTCTGGTTAGATGACACTCCTGCCCCAACTGCCACGAA
TCT**GGGGGGAGGCGGGGGTGGA**

MP1005 (tagging *NUD1* with *6Gly-3Flag::kanMX*; rev)
ATTTACTAATTACATACATTTTTAGTACTGCGTACGGGTATAGTTATG
GGG**GAATTCGAGCTCGTTTAAAC**

**Fluorescence microscopy**. F-actin staining was performed on cells fixed with 3.7% formaldehyde for 50 min under shaking at R.T. F-actin was visualized with Alexa Fluor 546-labeled phalloidin (A22283 Molecular Probes) at 20 U/ml after overnight incubation at 4 °C.

To detect spindle formation and elongation, alpha-tubulin immunostaining was performed on formaldehyde-fixed cells using the YOL34 monoclonal antibody (1:100; MCA78S AbD Serotec, Raleigh, NC), followed by indirect immunofluorescence using CY2-conjugated anti-rat antibody (1:100; 31645 Pierce Chemical Co.).

Detection of MEN factors at SPBs in anaphase was done in cells that were presynchronized in G1 and released in the appropriate medium for a sufficient time to enrich for anaphase cells (typically 90 and 105 min after release in YEPD and YEPRG, respectively). Cells were imaged after fixation with cold 100% ethanol. Fluorescence intensities in anaphase cells were measured with ImageJ on max-projected images (11 planes 0.3 μm spaced) after removing the background and applying a threshold that highlighted only SPB particles labeled by Spc42-mCherry. The selected region of interests (ROIs) were then used to measure fluorescence

intensities in the GFP channel with the ImageJ Analyze Particles tool. Min intensities were considered as cytoplasmic fluorescence, while max intensities corresponded the maximal pixel values within SPBs. Data were finally plotted according to the following equation: $(max_{GFP} - min_{GFP})/(max_{mCherry} - min_{mCherry})$. Buds and mother cells were distinguished on the basis of the alpha factor-induced shmoo-shaped morphology of mother cells.

Still digital images were taken with an oil immersion 100× 1.4 HCX Plan-Apochromat objective (Zeiss) with a Coolsnap HQ2 CDD camera (Photometrics) mounted on a Zeiss AxioimagerZ1 fluorescence microscope and controlled by the MetaMorph imaging system software. Z stacks containing 11 planes were acquired with a step size of 0.3 μm and a binning of 1. Z stacks were max-projected and calibrated using ImageJ.

For time-lapse video microscopy cells were mounted on 1% agarose pads in SD medium on Fluorodishes and filmed at controlled temperature with either a 100× 1.45 NA oil immersion objective mounted on a Spinning disk CSU-X1 Andor Nikon Eclipse Ti microscope coupled to an iXon Ultra camera controlled by the Andor iQ3 software (Figs. 1b, 2a, 2c, 3b, 4a–e and Supplementary Figs. 1a–d, 1f, 2a, 2c, d) or a 100× 1.49 NA oil immersion objective mounted on a Nikon Eclipse Ti microscope equipped with an EMCCD Evolve 512 Camera (Photometrics) and iLAS[2] module (Roper Scientific) and controlled by Metamorph (Figs. 2e, 6c, 7a, b and Supplementary Figs. 1e, 2e, 3a–d, 4a, b, 5a, b). Z stacks of 10–12 planes were acquired every 2–5 min with a step size of 0.4–0.5 μm and a binning of 1. Z stacks were max-projected with ImageJ or Metamorph.

**3D structural illumination microscopy (SIM)**. Cells were grown to log phase in YEPD at 25 °C, washed three times with BRB80 buffer (80 mM PIPES pH 6.9, 1 mM MgCl2, 1 mM EGTA), mounted on Concavalin A-coated coverslips and fixed with 4% formaldehyde for 15 min, followed by two washes with BRB80-NH4Cl (BRB80 with 50 mM NH4Cl). 3D-SIM was performed with an OMX DeltaVision equipped with Evolve 512 EMCCD cameras. After consulting the NYQUIST calculator to determine the ideal Voxel size, 25 Z-stacks of 0.125 μm thickness were acquired with a 100× Plan SuperApochromat 1.4 objective and an immersion oil with 1.519 refraction index. Raw data were processed with SoftWoRx and the quality of SIM-reconstructed images was assessed by the SIMcheck plugin of ImageJ.

**Quantification of septin and AMR-associated fluorescence intensities**. Fluorescence intensities associated to Shs1-mCherry and Myo1-GFP were quantified with ImageJ on sum-projected Z-stacks. Intensities were measured with the Time Series Analyser V3 plugin of ImageJ within a ROI defining the bud neck. For each channel the background was defined as the minimal intensity detected within the ROI during the movie and subtracted from fluorescence intensity values. The highest fluorescence intensity reached at any time point within the time frame of interest was set at 100%, while all the other fluorescence intensities were relative to this reference value. Relative intensities were then averaged after setting as time 0 the time immediately preceding septin ring splitting.

**Transmission electron microscopy**. Logarithmically growing cells were fixed with 3% paraformaldehyde and 0.5% glutaraldehyde in 0.1 M sodium phosphate buffer pH 7.4 (PB) for 2 h at room temperature, then washed three times with PB buffer and incubated in 1% of Na meta-periodate in 0.1 M PB at room temperature for 30 min. After washing, cells were resuspended in 50 mM NH4Cl. Cells were subjected to a gradient of ethanol before being embedded in LR White resin and let to polymerize for 2 days at 60 °C. Samples were sectioned at 100 nm, collected on 100 mesh copper grid supported with carbon/formvar (EMS, USA), and stained with Uranyl Acetate in MeOH for 4 min, followed by Sato's lead citrate for 2 min. Ultrathin sections were examined with a JEM-2200FS electron microscope at 100 kV equipped with a TemCam-F416 CMOS camera and controlled by the software package EM-MENU.

**Detection of ubiquitin conjugates**. Cells carrying a centromeric plasmid expressing ubiquitin or six His-tagged ubiquitin from the copper-inducible *CUP1* promoter were grown to log phase at 30 °C in selective medium. Ubiquitin expression was induced with 250 μM CuSO4 for 3 h. Cells were then harvested by centrifugation at 2000g and washed in cold water to be lysed in 1.85 M NaOH/7.5% β-mercaptoethanol (Fig. 5a, b, Supplementary Figs. 8 and 10b) for 20 min at 4 °C before precipitation with 10% trichloro acetic acid (TCA) at 4 °C. Alternatively, cells were resuspended in 10% TCA and lysed by mechanical shearing with glass beads (Fig. 5c and Supplementary Fig. 10a). TCA precipitates were resuspended in buffer A (6 M guanidium-Cl, 100 mM NaPO4 pH 8, 10 mM Tris-Cl pH 8) with shaking for at least 4 h, and the debris was removed by centrifugation at 2000g. Lysates were incubated overnight at room temperature with Ni-NTA agarose beads (Qiagen, Valencia, CA) in the presence of 15 mM imidazole and 0.05% Tween20. Beads were then washed twice with buffer A supplemented with 0.05% Tween 20 and three times with buffer C (8 M urea, 100 mM NaPO4 pH 6.3, 10 mM Tris-Cl pH 6.3, 0.05% Tween 20). Bound proteins were eluted by addition of 30 μl of HU buffer (8 M urea, 200 mM Tris-Cl pH 6.8, 1 mM EDTA, 5% SDS, 0.1% bromo-phenol blue, 1.5% dithiothreitol), heated 10 min at 65 °C and then subjected to

SDS-PAGE on precast gradient gels (4–15% BioRad) followed by western blot analysis.

**Protein extracts and western blotting**. TCA protein extracts were prepared as previously described[31] for western blot analysis. Briefly, 10–15 ml of cell culture in logarithmic phase ($OD_{600} = 05-1$) were collected by centrifugation at $2000g$, washed with 1 ml of 20% TCA and resuspended in 100 µl of 20% TCA before breakage of cells with glass beads (diameter 0.5–0.75 mm) on a Vibrax VXR (IKA). After addition of 400 µl of 5% TCA, lysates were centrifuged for 10 min at $845 g$. Protein precipitates were resuspended in 100 µl of 3× SDS sample buffer (240 mM Tris-Cl pH6.8, 6% SDS, 30% glycerol, 2.28 M β-mercaptoethanol, 0.06% bromophenol blue), denatured at 99 °C for 3 min and loaded on SDS-PAGE after elimination of cellular debris by centrifugation (5 min at $20,000g$).

Denaturing IPs to monitor Nud1-S78 phosphorylation were performed as described[16]. Briefly, 12.5 $OD_{600}$ units of cells were spun down at $2000g$ and resuspended in 5 ml of 5% TCA. After ≥10 min incubation in ice, cells were spun down at $2000g$, washed once with 1 ml of 50 mM Tris-Cl pH 7.5 and once with 1 ml of acetone. After acetone removal, cell pellets were dried overnight before cell breakage with glass beads at 4 °C in 180 µl of lysis buffer (50 mM Tris-Cl pH7.5, 1 mM EDTA pH 8, 5 mM DTT containing a cocktail of protease inhibitors (Complete EDTA-free Roche) and phosphatase inhibitors (PhosSTOP Roche)). After addition of 9 µl of 20% SDS lysates were heated at 100 °C for 5 min followed by addition of 1.71 ml of cold NP40 buffer (150 mM NaCl, 50 mM Tris-Cl pH 7.5, 1% NP40). Lysates were cleared at $20,000g$ for 10 min at 4 °C and incubated on a nutator for 2 h at 4 °C with 50 µl of protein A-sepharose pre-adsorbed with 2.5 µl of anti-PK antibody (MCA1360 AbD Serotec). The resin was spun down at 4 °C at $845 g$, washed three times with NP40 buffer and eluted with 30 µl of 3× SDS sample buffer (240 mM Tris-Cl pH6.8, 6% SDS, 30% glycerol, 2.28 M β-mercaptoethanol, 0.06% bromophenol blue) at 99 °C. Eluates were spun down at $20,000g$ and loaded on 8% SDS-PAGE.

Proteins were wet-transferred to Protran membranes (Schleicher and Schuell) overnight at 0.2 A and probed with monoclonal anti-HA 12CA5 (1:5000), anti-Flag M2 (F1804 Sigma Aldrich, 1:5000) or anti-PK (alias anti-V5; MCA1360 AbD Serotec, 1:3000) or polyclonal anti-Cdc5 (sc-6733 Santa Cruz, 1:3000), anti-Nud1-pS78 (a generous gift from A. Amon, 1:1000), anti-Clb2 (a generous gift from W. Zachariae, 1:2000) and anti-Spc72 (a generous gift from M. Knop, 1:1000) antibodies diluted in 5% low-fat milk (Regilait). Secondary antibodies were purchased from GE Healthcare and proteins were detected by a home-made enhanced chemiluminescence system. Uncropped blots are available in the Supplementary Information.

**FACS analysis of DNA contents**. For flow cytometric DNA quantification, $5 \times 10^6 - 2 \times 10^7$ cells were collected at each time point, spun at $10,000g$ and fixed with 1 ml of 70% ethanol for at least 30 min at RT. After one wash with 50 mM Tris-Cl pH 7.5, cells were resuspended in 0.5 ml of the same buffer containing 0.025 ml of a preboiled 10 mg/ml RNAse solution and incubated overnight at 37 °C. The next day cells were spun at $10,000g$ and resuspended in 0.5 ml of 5 mg/ml pepsin freshly diluted in in 55 mM HCl. After 30 min incubation at 37 °C cells were washed with FACS buffer (200 mM Tris pH 7.5, 200 mM NaCl, 78 mM $MgCl_2$) and resuspended in the same buffer containing 50 µg/ml propidium iodide. After a short sonication samples were diluted (1:20–1:10) in 1 ml of 50 mM Tris-Cl pH 7.5 and analyzed with a FACSCalibur device (BD Biosciences). Totally, 10,000 events were scored for each sample and plotted after gating out the debris as illustrated in Supplementary Fig. 12.

## Data availability

Data supporting the findings of this study are available in the article and Supplementary Information files, or from the corresponding author upon request.

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

## Acknowledgments

We are grateful to A. Amon, Y. Barral, E. Bi, E. Conibear, F. Cross, R. Deshaies, A. Fatica, D. Finley, E. Johnson, M. Kanemaki, J. Kilmartin, H. Leonhardt, F. Luca, G. Pereira, J. Pringle, D. Stillman, R. Visintin, and W. Zachariae for providing reagents; to N.V. Gounko at the IMB-IMCB Joint Electron Microscopy Suite for assistance with TEM specimen preparation; to members of Piatti's lab for useful discussions; to E. Schwob and A. Devault for critical reading of the manuscript. This work has been supported by the Fondation ARC for Cancer Research (grant PJA 20141201926 to S.P.) and the Merlion Exchange Program to S.P and G.R. D.T. was supported by an Erasmus Placement traineeship. We acknowledge the imaging facility MRI, member of the national infrastructure France-BioImaging supported by the French National Research Agency (ANR-10-INBS-04, "Investments for the future").

## Author contributions

S.P. conceived the project. D.T., M.A.J, S.I, G.R. and S.P. designed and carried out the experiments and analyzed the relative data. D.T., S.P. and M.A.J. made the figures. S.P. wrote the manuscript with inputs from all the authors.

## Additional information

**Competing interests:** The authors declare no competing interests.

