## [Peer Review File · Nature Communications]

Reviewers' comments:

Reviewer #1 (Remarks to the Author):

It was previously suggested that septin ring splitting may remove a structural barrier for the actomyosin ring to contract (e.g. Lippincott et al, 2001). Tamborrini et al. test this hypothesis. Briefly, the authors show that the components of the mitotic exit network (MEN) function in controlling septin remodeling at the division site during mitotic exit and that this remodeling is required for actomyosin ring constriction. They further link the E3 ubiquitin ligases Dma1 and Dma2 to regulation of MEN signaling at the SPBs through ubiquitination of the MEN scaffold Nud1. Finally, they show that constitutive Nud1-dependent anchoring of the phosphatase Cdc14 at the SPBs is sufficient for septin clearance from the division site.

I find the manuscript interesting. Having said that, I have a number of suggestions and questions that in my opinion should be addressed prior to publication.

What are the FACS profiles of strains shown in Figure 4? Is it possible to distinguish between myosin clearance from the bud neck and productive constriction/cytokinesis? Do other, late cytokinetic proteins such as Csh2 exhibit normal dynamics under these conditions? Related to this, perhaps the authors could comment on the fact that the rate of myosin clearance from the bud neck (that they use as a proxy for ring constriction) in Dma2-overexpressing *cdc12-1* cells is significantly delayed as compared with the wild type.

Why destabilizing septins or expressing a dominant active form of Tem1 in Dma2-overexpressing cells lead to the block in mitotic exit (Fig. 4E)?

I am a bit confused regarding Fig. 5C: does the input panel suggest that Nud1 abundance changes during the cell cycle, peaking at mitotic exit? I agree that it appears that Nud1 ubiquitination peaks at the same time point but it is not clear from this figure if the ratio between ubiquitinated and non-ubiquitinated Nud1 in fact changes in the cell cycle. Is this possible to quantitate?

The authors should provide examples of primary data summarized in Fig. 5D. Related to this set of data, given that Nud1 is thought to recruit Tem1 and the rest of the MEN cascade in a hierarchical manner, why Tem1 recruitment is not affected in Dma2 overexpressing cells? As a side note, it appears that Mob1 signal 'oscillates' between the two SPBs in DMA2-overexpressing cells (Fig. 5E). Perhaps the authors could comment on these observations.

It is well established in the field that once split, the septin 'rings' are separated considerably from the actomyosin ring, as also shown by the authors in Fig. 1. However, we know very little about actomyosin and septin organization prior to this event. Given that the authors suggest that septin collar may prevent ring constriction, I wonder if the authors could use their SIM protocol to provide high-resolution images of septin collar and the division ring before the septin rings are 'split', to complement their nice super-resolution images of constricting rings.

Is enzymatic activity of Cdc14 tethered to Nud1 at the SPBs required for its function in removing septins from the division site (Fig. 5)? Related to this set of data, it appears that Cdc14 recruitment to the SPBs leads to bulk septin clearance from the bud neck rather than the normal septin ring splitting. The authors should at least comment on this important distinction in their manuscript.

Minor comments:

It would make sense to add data on the *mob1-77* mutant to Fig. S1, to show it alongside other MEN mutants and complement Fig. 2.

Fig. 3D. It is difficult to judge actin recruitment to the rings using phalloidin that stains all actin structures in the cell. Is it possible to use a definitive marker of late actomyosin rings such as Iqg1?

Discussion, page 17, the sentence stating that 'activation of the Cdc14 phosphatase thereby leads to inactivation of mitotic CDKs'. Please edit for clarity - Cdc14 dephosphorylates mitotic CDK targets.

Page 9, not clear what the authors call 'cleavage furrow' in the case of the budding yeast cytokinesis.

I am not convinced if it is necessary to show a considerable amount of data on Dbf2 given that it was already shown that septin rings split in *dbf2* mutant cells (Oh et al, 2012). If the authors feel it is central to their story, they should address seeming differences between the phenotypes observed by the Bi lab and their study. For example, Oh et al argue that Dbf2 inactivation actually leads to delay in ring constriction, with cells exhibiting split septin rings but unconstricted division rings. It appeared that the myosin in that study gradually disappeared from unconstricting Chs2-labeled rings (somewhat similarly to the phenotype shown in Miller et al, 2012) rather than underwent normal constriction-related dynamics.

Reviewer #2 (Remarks to the Author):

Cytokinesis is the final step of cell division, which ends up with the physical separation of the daughter cells after completion of mitosis. Using *Saccharomyces cerevisiae* as a model, Tamborrini et al. aim to provide new insights into the mechanisms that regulate this process. More specifically, the authors re-evaluate whether and how the Mitotic Exit Network (MEN), a signaling pathway that promotes CDK inactivation and exit from mitosis in budding yeast, directly promotes cytokinesis independently of its global role in promoting the reversal of the phosphorylation status of CDK substrates, among which there are different factors whose de-phosphorylation is a prerequisite for cytokinesis to take place. In their manuscript, Tamborrini et al. demonstrate that septin ring splitting and displacement from the division site is required for the subsequent constriction of the contractile actomyosin ring (CAR) and the completion of cytokinesis, and provide evidences that support a direct role for the MEN in promoting these series of events. Furthermore, the authors propose that this function of the MEN in facilitating splitting of the septin ring is independent of its role in mitotic exit signaling, and uncover a novel mechanism by which cells regulate MEN function to prevent cytokinesis under adverse conditions.

Although the authors shed new light into the process, the fact that the MEN plays a role in cytokinesis that its independent of its mitotic exit signaling function has been previously well established by different laboratories. Similarly, a role of Dma2 in the control of septin ring stability and CAR contraction has also been previously shown, as well as the consequences that changes in the levels of this U3 ubiquitin ligase impose on these aspects of the cytokinesis process. Nonetheless, the novel Dma2-dependent mechanism proposed to down-regulate the MEN under unfavorable situations is an interesting result that could reveal a new way by which pivotal cell cycle events are coordinated to maintain genome stability. Although this new aspect of the regulation of cytokinesis could make the manuscript potentially suitable for publication in *Nature Communications*, I have a few concerns as to whether the results fully support the conclusions drawn. The experiments described by the authors are mostly well executed and presented, but important controls are missing for some experiments. Also, a more robust demonstration of the proposed new role of Dma2 in the regulation of the MEN would be necessary. Therefore, I consider that the manuscript is still too preliminary in its present form and would require further experimental support to grant its publication.

My main concerns about the results are the following:

1.- The examination of septin ring disassembly and CAR constriction in different MEN mutants and after overexpression of Dma2 (Figures 2, 3, 4, 6, S1, S2 and S3) would greatly benefit from a more quantitative analysis that allowed a better estimation of the extent of the cytokinesis defects. In most cases, the authors only show representative images of a movie that follows cell cycle progression for selected cells, sometimes also including the results from the FACS analysis of DNA content to evaluate mitotic exit. Reinforcing this analysis with, at least, a quantification of the percentage of unbudded, budded and re-budded cells in a synchronized culture would further strengthen the authors' conclusions.

2.- The authors indicate that overexpression of Dma2 prevents septin ring splitting and CAR constriction, but does not affect mitotic exit. However, there are several observations in the manuscript that suggest that exit from mitosis could be, in fact, affected under these conditions. As such, Cdc5 levels remain extremely high (Figure 5F), and Cdc14 release seems to be also affected (data not shown), although it is not clear to what extent. Thus, a more thorough examination of the effects of increased levels of Dma2 on cell cycle progression should be included. Besides from better defining the effects of the overexpression of Dma2 on Cdc14 release (both FEAR and MEN-dependent) and on the localization of the phosphatase to the SPBs by including a more quantitative analysis, the authors could also evaluate the percentages of metaphase and anaphase cells, as well as the levels of molecular markers such as Sic1, Pds1, or Clb2, to better define the timing of cell cycle entry, the metaphase-to-anaphase transition and mitotic exit under these conditions.

3.- The authors state that ubiquitination of Nud1 is markedly affected by deletion of both DMA1 and DMA2. However, the effect of the lack of Dma1 and Dma2 on Nud1 ubiquitination is hard to assess in Figure 5A, since the amount of protein in the pulldowns is much lower for *dma1Δ dma2Δ* cells than for the wild type. The quality of the results shown in Figure 5C could also be improved to better evaluate how Nud1 ubiquitination changes throughout the cell cycle, since very different amounts of proteins were loaded in the different time points shown. Furthermore, it would be interesting to analyze how this cell-cycle dependent pattern of ubiquitination is affected by changes in the expression of Dma1 and Dma2.

4.- The localization of Dma2 to the SPBs, as well as the changes in the ubiquitination of Nud1 as a consequence of alterations in the level of expression of this U3 ubiquitin ligase, are in agreement with Dma2 directly ubiquitinating Nud1. However, this possibility could be further explored and substantiated. As such, it would be relevant to analyze whether Nud1 and Dma1 or Dma2 can co-immunoprecipitate and, if so, whether their interaction is cell cycle-regulated.

5.- Overexpression of Dma2 does not affect Tem1 or Cdc5 localization to the SPBs. However, and surprisingly, loading of Bub2-Bfa1 on these structures is strongly prevented under these conditions (Figure 5D). Since localization of Tem1 has been shown to be dependent on that of Bub2-Bfa1 (Pereira et al. 2000, among others), the authors should at least comment on this apparent contradiction.

6.- Figure 5F shows the analysis of Nud1 and Spc72 phosphorylation in cells synchronously progressing through mitosis, and the changes in the phosphorylation status of these proteins caused by overexpression of Dma2. Evaluation of the results, however, is complicated by the fact that there is no indication of how the cells progressed throughout the cell cycle under these conditions. An analysis of the kinetics of cell budding, the percentage of metaphase and anaphase cells, and the levels of molecular markers for specific cell cycle transitions, would facilitate evaluation of the results. Also, and importantly, this experiment would greatly benefit from a more precise estimation of the effects of Dma2 overexpression on the levels of phosphorylated Nud1 and Spc72 (e.g., quantitative western blot analysis and use of a protein encoded by a housekeeping gene as a loading control).

7.- The authors postulate that it is a defective Cdc14 recruitment to the SPBs that is responsible for the defects in septin ring disassembly after overexpression of Dma2. In order to validate this hypothesis, they checked whether constitutive anchoring of Cdc14 to the SPBs could restore septin ring clearance from the division site in Dma2-overexpressing cells. However, the results from the FACS analysis in Figure 6 show a similar behavior for GAL1-DMA2 NUD1-GBD CDC14-GFP cells and the GAL1-DMA2 CDC14-GFP control. A more quantitative evaluation of the extent of the recovery of the cytokinesis defects in GAL1-DMA2 cells as a consequence of the constitutive targeting of Cdc14 to the SPBs should be included for all the strains in Figure 6C. Specifically, and as previously already indicated in point #1, it would be particularly relevant in this case to analyze the percentage of unbudded, budded and re-budded cells on synchronized cell populations of each strain in the presence of galactose.

8.- Since Bub2-Bfa1 play a key role in the recruitment of Cdc14 to the SPBs, an additional prediction from the hypothesis that defective Cdc14 recruitment to the SPBs is the leading cause for the cytokinesis defects in Dma2-overexpressing cells, is that *bfa1Δ* or *bub2Δ* cells should display defects in septin ring disassembly and CAR constriction. The authors could evaluate this possibility to give further support to their hypothesis.

9.- The analysis of septin ubiquitination in Figures S5A and S5B seems to surprisingly indicate that ubiquitination of Cdc11 and Shs1 is heavily increased in *dma1Δ dma2Δ* cells. This is not observed, however, neither for Nud1 (Figure 5A; note that here there is, in fact, a decrease in the levels of ubiquitinated protein, as expected) nor for Tem1 (Figure S5C). Could the authors comment on this observation?

10.- The fact that Dma1/2 localize on the SPBs in late mitosis is highly interesting, especially taking into account the results presented in the manuscript that suggest that these proteins ubiquitinate the SPB component Nud1, possibly to turn down MEN signaling after cytokinesis. The authors could analyze the effects of the constitutive targeting of Dma1/2 to the SPBs on cell cycle progression and/or cytokinesis. Since they have already generated a GBD-tagged allele of NUD1, and by expressing in these cells GFP-tagged Dma2, it is a pretty straightforward experiment to do that could be very informative.

11.- Different results shown in the manuscript contradict previously published data by Cassini et al. (Cell cycle, 2013). As such, and while the authors show that destabilization of the septin ring drives CAR constriction in cells that overexpress Dma2, Cassini et al. have previously suggested, also using the *cdc12-1* mutant, that defective actomyosin ring contraction in Dma2-overproducing cells is not caused by hyper-stabilization of the septin ring. More importantly, and in contrast to what shown in Figure S5D, they also showed that ubiquitination of Tem1 is enhanced after overexpression of Dma2. Furthermore, Cassani et al. suggested that Dma2 regulates cytokinesis by promoting ubiquitination of Tem1, which might inhibit Tem1 binding to Iqg1, whose association has been proposed to be an essential step for CAR contraction. The authors should at least comment on these results in the discussion, and try to fit this previously established function of Dma2 in the regulation of the MEN into their model.

Finally, some minor points are:

12.- The graphs in figures 3C, 3D, and 4E do not include error bars. These figures also lack the corresponding analyses to evaluate the statistical significance of the results.

13.- A black box shows up, probably by mistake, in the background of the graph shown in Figure 3C.

14.- The authors could complement the discussion in page 17 by further indicating why it is relevant for haploid budding yeast cells to establish an axial budding pattern.

Reviewer #3 (Remarks to the Author):

Review on „Recruitment of the Mitotic Exit Network to the yeast centrosome couples septin displacement to actomyosin ring constriction“ by Davide Tamborrini et al.

The manuscript addresses the important issue in yeast cell cycle and cytokinesis research, how septin ring splitting is coupled to acto-myosin-ring contraction.

Through a number of well designed and well performed experiments the authors arrive at the conclusion that components of the MEN, independently of their function in driving mitotic exit, promote septin ring splitting. A plausible model was derived in which the recruitment of MEN components to the SPB initiates a signal for ring splitting. The recruitment and the signal can be inhibited by ubiquitinylation of the SPB component Nud1. I recommend to accept the manuscript for publication.

Listed below are suggestions and points of critique the authors might consider to improve the manuscript:

1. Experiments of Fig. 2a, c, e need a quantitative statement on how often and robust the phenotypes were observed.
2. Page 12; Figure S5A, B: The more relevant experiment in this context is to show whether over expression of DMA2 influences the ubiquitinylation of Cdc11 and Shs1.
3. Page 12, Figure 5: Figure 5A: Lane 1 seems to suggest that Nud1-3PK is precipitated by Ni-NTA. If true, how does this influence the interpretation of the other pulldowns. Please comment.
4. Page 12, Figure 5: Figure 5A, B: The input lanes indicate that the great majority of Nud1-3PK is not ubiquitinated even under conditions of DMA2 over-expression. If this is true, how is MEN component-recruitment to the SPB be inhibited by Num1 ubiquitinylation when only a minority of Num1 gets modified? Please comment.
5. Page 12, Figure 5: Figure 5E: Please insert “Mob1-GFP” at the top of the time lapse for the ease of reading.
6. Page 15, Figure 6B: Upon artificial recruitment of Cdc14 to the SPB, the septin never splits but simply disappears. Do we still look at the same mechanism as in the wild type cells? The phenotype resembles the one observed in BUD4 deletion strains. Does it make sense to monitor the distribution of Bud4 under those conditions? Please comment and discuss.
7. Poly-ubiquitinylation seems not to affect the levels of Nud1 in the cell. This is unusual. Does it mean that ubiquitinylation of Nud1 blocks association with a critical component? If the authors have a MEN-component as candidate of a direct binding partner of Nud1, should this not be tested directly?
8. Materials and Methods: Please provide the protocol for the synchronization of the yeast cells. The PK tag is not that common. Please describe.
9. My impression is that the cartoon of the model is not informative enough to justify a dedicated Figure.

Reviewers' comments:

Reviewer #1 (Remarks to the Author):

It was previously suggested that septin ring splitting may remove a structural barrier for the actomyosin ring to contract (e.g. Lippincott et al, 2001). Tamborrini et al. test this hypothesis. Briefly, the authors show that the components of the mitotic exit network (MEN) function in controlling septin remodeling at the division site during mitotic exit and that this remodeling is required for actomyosin ring constriction. They further link the E3 ubiquitin ligases Dma1 and Dma2 to regulation of MEN signaling at the SPBs through ubiquitination of the MEN scaffold Nud1. Finally, they show that constitutive Nud1-dependent anchoring of the phosphatase Cdc14 at the SPBs is sufficient for septin clearance from the division site.

I find the manuscript interesting. Having said that, I have a number of suggestions and questions that in my opinion should be addressed prior to publication.

What are the FACS profiles of strains shown in Figure 4? Is it possible to distinguish between myosin clearance from the bud neck and productive constriction/cytokinesis? Do other, late cytokinetic proteins such as Csh2 exhibit normal dynamics under these conditions? Related to this, perhaps the authors could comment on the fact that the rate of myosin clearance from the bud neck (that they use as a proxy for ring constriction) in Dma2-overexpressing *cdc12-1* cells is significantly delayed as compared with the wild type.

Why destabilizing septins or expressing a dominant active form of Tem1 in Dma2-overexpressing cells lead to the block in mitotic exit (Fig. 4E)?

I am a bit confused regarding Fig. 5C: does the input panel suggest that Nud1 abundance changes during the cell cycle, peaking at mitotic exit? I agree that it appears that Nud1 ubiquitination peaks at the same time point but it is not clear from this figure if the ratio between ubiquitinated and non-ubiquitinated Nud1 in fact changes in the cell cycle. Is this possible to quantitate?

The authors should provide examples of primary data summarized in Fig. 5D. Related to this set of data, given that Nud1 is thought to recruit Tem1 and the rest of the MEN cascade in a hierarchical manner, why Tem1 recruitment is not affected in Dma2 overexpressing cells? As a side note, it appears that Mob1 signal 'oscillates' between the two SPBs in DMA2-overexpressing cells (Fig. 5E). Perhaps the authors could comment on these observations.

It is well established in the field that once split, the septin 'rings' are separated considerably from the actomyosin ring, as also shown by the authors in Fig. 1. However, we know very little about actomyosin and septin organization prior to this event. Given that the authors suggest that septin collar may prevent ring constriction, I wonder if the authors could use their SIM protocol to provide high-resolution images of septin collar and the division ring before the septin rings are 'split', to complement their nice super-resolution images of constricting rings.

Is enzymatic activity of Cdc14 tethered to Nud1 at the SPBs required for its function in removing septins from the division site (Fig. 5)? Related to this set of data, it appears that Cdc14 recruitment to the SPBs leads to bulk septin clearance from the bud neck rather than the normal septin ring splitting. The authors should at least comment on this important distinction in their manuscript.

Minor comments:

It would make sense to add data on the *mob1-77* mutant to Fig. S1, to show it alongside other MEN mutants and complement Fig. 2.

Fig. 3D. It is difficult to judge actin recruitment to the rings using phalloidin that stains all actin structures in the cell. Is it possible to use a definitive marker of late actomyosin rings such as Iqg1?

Discussion, page 17, the sentence stating that 'activation of the Cdc14 phosphatase thereby leads to inactivation of mitotic CDKs'. Please edit for clarity - Cdc14 dephosphorylates mitotic CDK targets.

Page 9, not clear what the authors call 'cleavage furrow' in the case of the budding yeast cytokinesis.

I am not convinced if it is necessary to show a considerable amount of data on Dbf2 given that it was already shown that septin rings split in *dbf2* mutant cells (Oh et al, 2012). If the authors feel it is central to their story, they should address seeming differences between the phenotypes observed by the Bi lab and their study. For example, Oh et al argue that Dbf2 inactivation actually leads to delay in ring constriction, with cells exhibiting split septin rings but unconstricted division rings. It appeared that the myosin in that study gradually disappeared from unconstricting Chs2-labeled rings (somewhat similarly to the phenotype shown in Miller et al, 2012) rather than underwent normal constriction-related dynamics.

Reviewer #2 (Remarks to the Author):

Cytokinesis is the final step of cell division, which ends up with the physical separation of the daughter cells after completion of mitosis. Using *Saccharomyces cerevisiae* as a model, Tamborrini et al. aim to provide new insights into the mechanisms that regulate this process. More specifically, the authors re-evaluate whether and how the Mitotic Exit Network (MEN), a signaling pathway that promotes CDK inactivation and exit from mitosis in budding yeast, directly promotes cytokinesis independently of its global role in promoting the reversal of the phosphorylation status of CDK substrates, among which there are different factors whose de-phosphorylation is a prerequisite for cytokinesis to take place. In their manuscript, Tamborrini et al. demonstrate that septin ring splitting and displacement from the division site is required for the subsequent constriction of the contractile actomyosin ring (CAR) and the completion of cytokinesis, and provide evidences that support a direct role for the MEN in promoting these series of events. Furthermore, the authors propose that this function of the MEN in facilitating splitting of the septin ring is independent of its role in mitotic exit signaling, and uncover a novel mechanism by which cells regulate MEN function to prevent cytokinesis under adverse conditions.

Although the authors shed new light into the process, the fact that the MEN plays a role in cytokinesis that its independent of its mitotic exit signaling function has been previously well established by different laboratories. Similarly, a role of Dma2 in the control of septin ring stability and CAR contraction has also been previously shown, as well as the consequences that changes in the levels of this U3 ubiquitin ligase impose on these aspects of the cytokinesis process. Nonetheless, the novel Dma2-dependent mechanism proposed to down-regulate the MEN under unfavorable situations is an interesting result that could reveal a new way by which pivotal cell cycle events are coordinated to maintain genome stability. Although this new aspect of the regulation of cytokinesis could make the manuscript potentially suitable for publication in *Nature Communications*, I have a few concerns as to whether the results fully support the conclusions drawn. The experiments described by the authors are mostly well executed and presented, but important controls are missing for some experiments. Also, a more robust demonstration of the proposed new role of Dma2 in the regulation of the MEN would be necessary. Therefore, I consider that the manuscript is still too preliminary in its present form and would require further experimental support to grant its publication.

My main concerns about the results are the following:

1.- The examination of septin ring disassembly and CAR constriction in different MEN mutants and after overexpression of Dma2 (Figures 2, 3, 4, 6, S1, S2 and S3) would greatly benefit from a more quantitative analysis that allowed a better estimation of the extent of the cytokinesis defects. In most cases, the authors only show representative images of a movie that follows cell cycle progression for selected cells, sometimes also including the results from the FACS analysis of DNA content to evaluate mitotic exit. Reinforcing this analysis with, at least, a quantification of the percentage of unbudded, budded and re-budded cells in a synchronized culture would further strengthen the authors' conclusions.

2.- The authors indicate that overexpression of Dma2 prevents septin ring splitting and CAR constriction, but does not affect mitotic exit. However, there are several observations in the manuscript that suggest that exit from mitosis could be, in fact, affected under these conditions. As such, Cdc5 levels remain extremely high (Figure 5F), and Cdc14 release seems to be also affected (data not shown), although it is not clear to what extent. Thus, a more thorough examination of the effects of increased levels of Dma2 on cell cycle progression should be included. Besides from better defining the effects of the overexpression of Dma2 on Cdc14 release (both FEAR and MEN-dependent) and on the localization of the phosphatase to the SPBs by including a more quantitative analysis, the authors could also evaluate the percentages of metaphase and anaphase cells, as well as the levels of molecular markers such as Sic1, Pds1, or Clb2, to better define the timing of cell cycle entry, the metaphase-to-anaphase transition and mitotic exit under these conditions.

3.- The authors state that ubiquitination of Nud1 is markedly affected by deletion of both DMA1 and DMA2. However, the effect of the lack of Dma1 and Dma2 on Nud1 ubiquitination is hard to assess in Figure 5A, since the amount of protein in the pulldowns is much lower for *dma1Δ dma2Δ* cells than for the wild type. The quality of the results shown in Figure 5C could also be improved to better evaluate how Nud1 ubiquitination changes throughout the cell cycle, since very different amounts of proteins were loaded in the different time points shown. Furthermore, it would be interesting to analyze how this cell-cycle dependent pattern of ubiquitination is affected by changes in the expression of Dma1 and Dma2.

4.- The localization of Dma2 to the SPBs, as well as the changes in the ubiquitination of Nud1 as a consequence of alterations in the level of expression of this U3 ubiquitin ligase, are in agreement with Dma2 directly ubiquitinating Nud1. However, this possibility could be further explored and substantiated. As such, it would be relevant to analyze whether Nud1 and Dma1 or Dma2 can co-immunoprecipitate and, if so, whether their interaction is cell cycle-regulated.

5.- Overexpression of Dma2 does not affect Tem1 or Cdc5 localization to the SPBs. However, and surprisingly, loading of Bub2-Bfa1 on these structures is strongly prevented under these conditions (Figure 5D). Since localization of Tem1 has been shown to be dependent on that of Bub2-Bfa1 (Pereira et al. 2000, among others), the authors should at least comment on this apparent contradiction.

6.- Figure 5F shows the analysis of Nud1 and Spc72 phosphorylation in cells synchronously progressing through mitosis, and the changes in the phosphorylation status of these proteins caused by overexpression of Dma2. Evaluation of the results, however, is complicated by the fact that there is no indication of how the cells progressed throughout the cell cycle under these conditions. An analysis of the kinetics of cell budding, the percentage of metaphase and anaphase cells, and the levels of molecular markers for specific cell cycle transitions, would facilitate evaluation of the results. Also, and importantly, this experiment would greatly benefit from a more precise estimation of the effects of Dma2 overexpression on the levels of phosphorylated Nud1 and Spc72 (e.g., quantitative western blot analysis and use of a protein encoded by a housekeeping gene as a loading control).

7.- The authors postulate that it is a defective Cdc14 recruitment to the SPBs that is responsible for the defects in septin ring disassembly after overexpression of Dma2. In order to validate this hypothesis, they checked whether constitutive anchoring of Cdc14 to the SPBs could restore septin ring clearance from the division site in Dma2-overexpressing cells. However, the results from the FACS analysis in Figure 6 show a similar behavior for GAL1-DMA2 NUD1-GBD CDC14-GFP cells and the GAL1-DMA2 CDC14-GFP control. A more quantitative evaluation of the extent of the recovery of the cytokinesis defects in GAL1-DMA2 cells as a consequence of the constitutive targeting of Cdc14 to the SPBs should be included for all the strains in Figure 6C. Specifically, and as previously already indicated in point #1, it would be particularly relevant in this case to analyze the percentage of unbudded, budded and re-budded cells on synchronized cell populations of each strain in the presence of galactose.

8.- Since Bub2-Bfa1 play a key role in the recruitment of Cdc14 to the SPBs, an additional prediction from the hypothesis that defective Cdc14 recruitment to the SPBs is the leading cause for the cytokinesis defects in Dma2-overexpressing cells, is that *bfa1Δ* or *bub2Δ* cells should display defects in septin ring disassembly and CAR constriction. The authors could evaluate this possibility to give further support to their hypothesis.

9.- The analysis of septin ubiquitination in Figures S5A and S5B seems to surprisingly indicate that ubiquitination of Cdc11 and Shs1 is heavily increased in *dma1Δ dma2Δ* cells. This is not observed, however, neither for Nud1 (Figure 5A; note that here there is, in fact, a decrease in the levels of ubiquitinated protein, as expected) nor for Tem1 (Figure S5C). Could the authors comment on this observation?

10.- The fact that Dma1/2 localize on the SPBs in late mitosis is highly interesting, especially taking into account the results presented in the manuscript that suggest that these proteins ubiquitinate the SPB component Nud1, possibly to turn down MEN signaling after cytokinesis. The authors could analyze the effects of the constitutive targeting of Dma1/2 to the SPBs on cell cycle progression and/or cytokinesis. Since they have already generated a GBD-tagged allele of NUD1, and by expressing in these cells GFP-tagged Dma2, it is a pretty straightforward experiment to do that could be very informative.

11.- Different results shown in the manuscript contradict previously published data by Cassini et al. (Cell cycle, 2013). As such, and while the authors show that destabilization of the septin ring drives CAR constriction in cells that overexpress Dma2, Cassini et al. have previously suggested, also using the *cdc12-1* mutant, that defective actomyosin ring contraction in Dma2-overproducing cells is not caused by hyper-stabilization of the septin ring. More importantly, and in contrast to what shown in Figure S5D, they also showed that ubiquitination of Tem1 is enhanced after overexpression of Dma2. Furthermore, Cassini et al. suggested that Dma2 regulates cytokinesis by promoting ubiquitination of Tem1, which might inhibit Tem1 binding to Iqg1, whose association has been proposed to be an essential step for CAR contraction. The authors should at least comment on these results in the discussion, and try to fit this previously established function of Dma2 in the regulation of the MEN into their model.

Finally, some minor points are:

12.- The graphs in figures 3C, 3D, and 4E do not include error bars. These figures also lack the corresponding analyses to evaluate the statistical significance of the results.

13.- A black box shows up, probably by mistake, in the background of the graph shown in Figure 3C.

14.- The authors could complement the discussion in page 17 by further indicating why it is relevant for haploid budding yeast cells to establish an axial budding pattern.

Reviewer #3 (Remarks to the Author):

Review on „Recruitment of the Mitotic Exit Network to the yeast centrosome couples septin displacement to actomyosin ring constriction“ by Davide Tamborrini et al.

The manuscript addresses the important issue in yeast cell cycle and cytokinesis research, how septin ring splitting is coupled to acto-myosin-ring contraction.

Through a number of well designed and well performed experiments the authors arrive at the conclusion that components of the MEN, independently of their function in driving mitotic exit, promote septin ring splitting. A plausible model was derived in which the recruitment of MEN components to the SPB initiates a signal for ring splitting. The recruitment and the signal can be inhibited by ubiquitinylation of the SPB component Nud1. I recommend to accept the manuscript for publication.

Listed below are suggestions and points of critique the authors might consider to improve the manuscript:

1. Experiments of Fig. 2a, c, e need a quantitative statement on how often and robust the phenotypes were observed.
2. Page 12; Figure S5A, B: The more relevant experiment in this context is to show whether over expression of DMA2 influences the ubiquitinylation of Cdc11 and Shs1.
3. Page 12, Figure 5: Figure 5A: Lane 1 seems to suggest that Nud1-3PK is precipitated by Ni-NTA. If true, how does this influence the interpretation of the other pulldowns. Please comment.
4. Page 12, Figure 5: Figure 5A, B: The input lanes indicate that the great majority of Nud1-3PK is not ubiquitinated even under conditions of DMA2 over-expression. If this is true, how is MEN component-recruitment to the SPB be inhibited by Num1 ubiquitinylation when only a minority of Num1 gets modified? Please comment.
5. Page 12, Figure 5: Figure 5E: Please insert “Mob1-GFP” at the top of the time lapse for the ease of reading.
6. Page 15, Figure 6B: Upon artificial recruitment of Cdc14 to the SPB, the septin never splits but simply disappears. Do we still look at the same mechanism as in the wild type cells? The phenotype resembles the one observed in BUD4 deletion strains. Does it make sense to monitor the distribution of Bud4 under those conditions? Please comment and discuss.
7. Poly-ubiquitinylation seems not to affect the levels of Nud1 in the cell. This is unusual. Does it mean that ubiquitinylation of Nud1 blocks association with a critical component? If the authors have a MEN-component as candidate of a direct binding partner of Nud1, should this not be tested directly?
8. Materials and Methods: Please provide the protocol for the synchronization of the yeast cells. The PK tag is not that common. Please describe.
9. My impression is that the cartoon of the model is not informative enough to justify a dedicated Figure.

Reviewers' comments:

Reviewer #1 (Remarks to the Author):

The authors have addressed my comments in full and I support publication of this manuscript. I have a few minor suggestions:

Page 5. I am not sure what the sentence "Ubiquitination of the MEN scaffold Nud1 at SPBs prevents septin splitting and CAR contraction to silence these processes at the end of cytokinesis" means. Presumably at the end of cytokinesis there is no need to silence these processes as they've already occurred? Perhaps this can be rephrased.

Page 10. Related to Iqg1 being 'slowly degraded' (Fig. S4). I am not sure one can conclude it is degraded, in the absence of appropriate measurements. The only thing I can conclude from the presented time-lapse is that Iqg1 disappears from the ring.

Page 13. In a sentence starting with 'In spite of their apparently normal cytokinesis, GAL1-DMA2 TEM1-Q79L cells could not complete cell division..', I think a more appropriate way to state it would be 'In spite of their apparently normal ring constriction'.

Reviewer #2 (Remarks to the Author):

In the revised version of their manuscript, Tamborrini et al. have made a significant effort to answer to the reviewer's concerns. The authors have now included additional data that further support some of their claims, especially those regarding the role of Dma2 as an inhibitor of the role of the Mitotic Exit Network (MEN) in promoting cytokinesis. In this regard, the inhibition of cytokinesis by the constitutive targeting of Dma1/2 to the SPBs is a strong new argument favoring their hypothesis. However, in my opinion, the proposed mechanism by which Dma1/2 regulates the cytokinesis-promoting function of the MEN still falls somewhat short of strong experimental support. I am particularly concerned about whether the data in the manuscript satisfactorily proves that the Dma1/2-dependent inhibition of the MEN is mediated by direct interaction of Dma1/2 with Nud1 and ubiquitination of this SPB-component by the E3 ubiquitin ligase. Two main weak points related to this issue are the following:

1.- As previously raised to the consideration of the authors, one would expect that if Dma1/2 directly ubiquitinated Nud1 these proteins should interact with each other. Tamborrini et al. have made an effort to test this possibility by carrying out co-immunoprecipitation experiments with cells co-expressing Nud1-3PK and Dma2-3HA that, unfortunately, have been inconclusive due to unspecific binding of Dma2 to the beads. I believe, however, that this is an important issue that deserved a further effort. Not only the authors could have tried other tagged versions of the proteins, which they already have available, but there are additional ways to test this interaction that could even provide further information about it. An interesting option that the authors could explore is to use the Bimolecular Fluorescence Complementation Assay (Sung et al., 2007), which could not only confirm the interaction but also indicate where this interaction occurs within the cell (in this case, hopefully in the context of the SPB).

2.- With regards to the analysis of post-translational modifications, there is a general inconsistency in the levels of Nud1 protein both in the inputs and in the pulldowns that makes it difficult to draw solid conclusions from the results shown. A reliable quantification of the fraction of ubiquitinated versus non-ubiquitinated Nud1, which the authors admit in their response to Reviewer #1 that is difficult to be estimated in their assays, would be required. One main problem seems to be the in vitro degradation of Nud1 in the conditions used for these assays, which is not however observed when protein extracts were prepared using TCA. If this were the case, they could simply use alternative extraction protocols in order to avoid this issue also in the pull-down experiments. As indicated in my original review, I believe that it would be critical to precisely determine both the

pattern of Nud1 ubiquitination throughout the cell cycle and how the changes in the expression of Dma1/2 modify this pattern. I do not coincide with the authors in that it should be tricky to evaluate this effect on a synchronized time course experiment (levels of Dma2 overexpression can be easily evaluated during the experiment for comparison), and the new data provided in Figure S9A is again difficult to interpret as a consequence of the differences in the levels of Nud1 protein among the different conditions and also among the inputs and the pull-downs.

Therefore, and despite still believing that the conclusions from this manuscript are potentially of interest in the field, I consider that, ideally, clarification of these two fundamental issues regarding the molecular mechanism proposed for Dma1/2-mediated inhibition of MEN signaling should be recommended before granting its final publication. Finally, some minor comments about the manuscript are:

- 1.- The authors claim to have fixed the problem regarding the black box that showed up in the background of Figure 3C from the original manuscript. However, the black box that obstructs the visibility of the graph shown in Figure 3C still shows up in the pdf file from the revised manuscript.
- 2.- In the text it is stated that introducing the TAB6-1 allele in GAL1-DMA2 cells accelerated mitotic exit (page 10 of the revised manuscript). However, no cell cycle progression analysis (or bibliographic reference) is provided.
- 3.- In page 13, the authors indicate that "deletion of both DMA1 and DMA2 [...] did not affect the ubiquitination pattern of either Cdc11 or Shs1 (Fig. S8A, B)". However, and as admitted in their reply to one of my concerns, ubiquitination of both septins is, in fact, heavily increased by the simultaneous lack of Dma1 and Dma2. The sentence should be thus corrected, and this observation commented in the manuscript.
- 4.- There are some overstatements in the description of the results. In this sense, in page 15 it is said that "Localization of Bub2-Bfa1, Cdc15 and Mob1 at SPBs was markedly inhibited...". Similarly, in page 16 it is stated that "[Total] Nud1 phosphorylation was markedly impaired upon DMA2-overexpression". In both cases, changes are subtler than what these sentences imply.

Reviewer #3 (Remarks to the Author):

Review on „Recruitment of the Mitotic Exit Network to the yeast centrosome couples septin displacement to actomyosin ring constriction“ by Davide Tamborrini et al.

I found the response of the authors to my suggestions and criticisms satisfying. I support the publication of the manuscript in Nature Communication.

Reviewer #1:

Page 5. I am not sure what the sentence "Ubiquitination of the MEN scaffold Nud1 at SPBs prevents septin splitting and CAR contraction to silence these processes at the end of cytokinesis" means. Presumably at the end of cytokinesis there is no need to silence these processes as they've already occurred? Perhaps this can be rephrased.

The sentence was rephrased accordingly.

Page 10. Related to Iqg1 being 'slowly degraded' (Fig. S4). I am not sure one can conclude it is degraded, in the absence of appropriate measurements. The only thing I can conclude from the presented time-lapse is that Iqg1 disappears from the ring.

We agree and rephrased the sentence.

Page 13. In a sentence starting with 'In spite of their apparently normal cytokinesis, GAL1-DMA2 TEM1-Q79L cells could not complete cell division..', I think a more appropriate way to state it would be 'In spite of their apparently normal ring constriction'.

We agree and rephrased the sentence.

Reviewer #2:

1.- As previously raised to the consideration of the authors, one would expect that if Dma1/2 directly ubiquitinated Nud1 these proteins should interact with each other. Tamborrini et al. have made an effort to test this possibility by carrying out co-immunoprecipitation experiments with cells co-expressing Nud1-3PK and Dma2-3HA that, unfortunately, have been inconclusive due to unspecific binding of Dma2 to the beads. I believe, however, that this is an important issue that deserved a further effort. Not only the authors could have tried other tagged versions of the proteins, which they already have available, but there are additional ways to test this interaction that could even provide further information about it. An interesting option that the authors could explore is to use the Bimolecular Fluorescence Complementation Assay (Sung et al., 2007), which could not only confirm the interaction but also indicate where this interaction occurs within the cell (in this case, hopefully in the context of the SPB).

We have made a huge effort to find good conditions to probe the Nud1-Dma2 interaction by co-immunoprecipitation because Dma2 is particularly sticky and binds aspecifically to resins. We obtained the best results by immunoprecipitation of 3Flag-tagged Nud1 and elution of immunoprecipitates with an excess of 3XFlag peptide. Using this strategy we find that a small fraction of Dma2-3HA associates to Nud1-3Flag in anaphase (new Fig. S9).

The BiFC complementation assay proposed by the Reviewer as an alternative to co-IPs would have been in principle a good suggestion, but is known to frequently generate false positives, prompting the need for a proper control where one of the two binding partners carries a mutation in the binding interface (which we obviously do not know). Furthermore, slow maturation of the chromophore is considered another limiting factor for the visualization by BiFC of transient or dynamic interactions in cells (reviewed in

Kodama and Hu, 2012, *BioTechniques* 53 : 285 ; Miller et al., 2015, *J. Mol. Biol.* 427 : 2039).

In this context, it is also worth noting that although the Dma1 ubiquitin ligase (paralogue of Dma2) was found at SPBs in late mitosis (Yau et al., 2014), we could not detect Dma1 or Dma2 localised at SPBs in our yeast strain background, using either the published constructs or GFP-tagged variants that we made in our lab.

2.- With regards to the analysis of post-translational modifications, there is a general inconsistency in the levels of Nud1 protein both in the inputs and in the pull-downs that makes it difficult to draw solid conclusions from the results shown. A reliable quantification of the fraction of ubiquitinated versus non-ubiquitinated Nud1, which the authors admit in their response to Reviewer #1 that is difficult to be estimated in their assays, would be required. One main problem seems to be the in vitro degradation of Nud1 in the conditions used for these assays, which is not however observed when protein extracts were prepared using TCA. If this were the case, they could simply use alternative extraction protocols in order to avoid this issue also in the pull-down experiments. As indicated in my original review, I believe that it would be critical to precisely determine both the pattern of Nud1 ubiquitination throughout the cell cycle and how the changes in the expression of Dma1/2 modify this pattern. I do not coincide with the authors in that it should be tricky to evaluate this effect on a synchronized time course experiment (levels of Dma2 overexpression can be easily evaluated during the experiment for comparison), and the new data provided in Figure S9A is again difficult to interpret as a consequence of the differences in the levels of Nud1 protein among the different conditions and also among the inputs and the pull-downs.

Following the Reviewer's advice, we have spent a considerable amount of time to set up the conditions to preserve Nud1 protein stability in vitro in our ubiquitination assays. We have succeeded by lysing cells directly in TCA, as detailed in Materials and Methods. We have repeated our analysis of Nud1 ubiquitination throughout the cell cycle and our new data (Fig. 5c) show that Nud1 is ubiquitinated in late mitosis and in the following G1 phase.

Using these new conditions we have also performed Nud1 ubiquitination assays during a synchronous release of wild type versus *GAL1-DMA2* cells in the presence of galactose. These data show that *DMA2* overexpression can stimulate Nud1 ubiquitination throughout the cell cycle but most markedly in late mitosis and in G1, i.e. during the cell cycle phases when Nud1 ubiquitination reaches its maximal levels in wild type cells (new Fig. S10a). It should be noticed, however, that persistent *DMA2* overexpression combined with ubiquitin overexpression led unexpectedly to abnormal Nud1 destabilisation and accumulation of cells in mitosis. This is not what we see upon *DMA2* overexpression in cells expressing endogenous levels of ubiquitin (see for instance Fig. 5f and S11c).

Quantifying the fraction of ubiquitinated versus non-ubiquitinated Nud1 is technically impossible at this stage because it would require detection of upshifted ubiquitinated forms of Nud1 in the inputs, something that we have never detected so far. This argues that either only a small fraction of Nud1 is ubiquitinated (but biologically relevant!) or a fraction of total Nud1 goes to SPBs where it is ubiquitinated. Another major shortcoming of these measurements is that other post-translational modifications could upshift the

electrophoretic mobility of Nud1 indistinguishably from ubiquitination, thereby interfering with reliable measurements.

Minor points:

1.- The authors claim to have fixed the problem regarding the black box that showed up in the background of Figure 3C from the original manuscript. However, the black box that obstructs the visibility of the graph shown in Figure 3C still shows up in the pdf file from the revised manuscript.

We have tried to remake this figure with Inkscape and on some, but not all computers, the black box mentioned by the Reviewer remains apparent. We do not understand why this is the case, but it is likely due to the conversion from .svg to .pdf. We can provide the figure in .svg if required.

2.- In the text it is stated that introducing the TAB6-1 allele in GAL1-DMA2 cells accelerated mitotic exit (page 10 of the revised manuscript). However, no cell cycle progression analysis (or bibliographic reference) is provided.

Since the *TAB6-1* allele increases the fraction of *GAL1-DMA2* cells with more than one septin ring during a 2hr time frame, presumably it facilitates (« accelerates ») mitotic exit of *GAL1-DMA2* cells. We have rephrased this sentence for the sake of clarity.

3.- In page 13, the authors indicate that “deletion of both DMA1 and DMA2 [...] did not affect the ubiquitination pattern of either Cdc11 or Shs1 (Fig. S8A, B)”. However, and as admitted in their reply to one of my concerns, ubiquitination of both septins is, in fact, heavily increased by the simultaneous lack of Dma1 and Dma2. The sentence should be thus corrected, and this observation commented in the manuscript.

We have corrected this sentence, but at the moment we have no data to support any additional comment.

4.- There are some overstatements in the description of the results. In this sense, in page15 it is said that “Localization of Bub2-Bfa1, Cdc15 and Mob1 at SPBs was markedly inhibited...”. Similarly, in page 16 it is stated that “[Total] Nud1 phosphorylation was markedly impaired upon DMA2-overexpression”. In both cases, changes are subtler than what these sentences imply.

We have corrected these sentences accordingly.

REVIEWERS' COMMENTS:

Reviewer #1 (Remarks to the Author):

No further comments

Reviewer #2 (Remarks to the Author):

The new results incorporated by Tamborrini et al. in their last revised version significantly strengthen the main weak points that I raised in my comments, adding further experimental support to the conclusions drawn in their manuscript. Hence, I support its final acceptance in Nature Communications.

Note: Please, check whether reference to figure S10A in line 356 from page 15 is correct, since I believe it is a typo.

Reviewer #3 (Remarks to the Author):

No further comments

REVIEWERS' COMMENTS:

Reviewer #2 (Remarks to the Author):

The new results incorporated by Tamborrini et al. in their last revised version significantly strengthen the main weak points that I raised in my comments, adding further experimental support to the conclusions drawn in their manuscript. Hence, I support its final acceptance in Nature Communications.

Note: Please, check whether reference to figure S10A in line 356 from page 15 is correct, since I believe it is a typo.

Thank you for noticing this typo. We have now corrected it.

Sincerely

Simonetta Piatti